Report

# Structure of the [Ca]E2P intermediate of Ca²⁺-ATPase 1 from *Listeria monocytogenes*

Sara Basse Hansen[1,2], Rasmus Kock Flygaard [1,2], Magnus Kjaergaard[1,2,3] & Poul Nissen [1,2,3✉]

## Abstract

Active transport by P-type Ca²⁺-ATPases maintain internal calcium stores and a low cytosolic calcium concentration. Structural studies of mammalian sarco/endoplasmic reticulum Ca²⁺-ATPases (SERCA) have revealed several steps of the transport cycle, but a calcium-releasing intermediate has remained elusive. Single-molecule FRET studies of the bacterial Ca²⁺-ATPase LMCA1 revealed an intermediate of the transition between so-called [Ca]E1P and E2P states and suggested that calcium release from this intermediate was the essentially irreversible step of transport. Here, we present a 3.5 Å resolution cryo-EM structure for a four-glycine insertion mutant of LMCA1 in a lipid nanodisc obtained under conditions with calcium and ATP and adopting such an intermediate state, denoted [Ca]E2P. The cytosolic domains are positioned in the E2P-like conformation, while the calcium-binding transmembrane (TM) domain adopts a calcium-bound E1P-ADP-like conformation. Missing density for the E292 residue at the calcium site (the equivalent of SERCA1a E309) suggests flexibility and a site poised for calcium release and proton uptake. The structure suggests a mechanism where ADP release and re-organization of the cytoplasmic domains precede calcium release.

**Keywords** P-type ATPase; Ca2+-ATPase; Calcium Transport; Calcium-bound Intermediate
**Subject Categories** Membranes & Trafficking; Structural Biology

## Introduction

Cell viability depends on the interplay between many different pathways that are regulated and interconnected by e.g., metabolic or cell cycle feedback and environmental conditions. Changes in calcium concentration form potent signals in nearly all cells, and it is crucial to understand how calcium homeostasis is maintained and regulated. Ca²⁺-ATPases transport calcium out of the cell or into internal stores and maintain a low concentration of free cytoplasmic calcium that can be four orders of magnitude below surroundings (Carafoli, 2002; Dominguez, 2004). Considering also a typically negative membrane potential and the divalent charge of calcium ions, cell membrane calcium gradients are some of the most powerful electrochemical gradients known for biological systems. As a consequence, transport against such gradients by Ca²⁺-ATPases must include a practically irreversible step to avoid reflux. Structural and mechanistic examination of this irreversible step has remained elusive, but could explain key principles of active transporters.

The P-type ATPase family encompasses integral membrane proteins that share both structural and mechanistic features and large domain re-arrangements during the transport cycle (Dyla et al, 2019b). P-type ATPases perform their active transport with energy from ATP hydrolysis via autoformation and breakdown of a phosphoenzyme (Pedersen and Carafoli, 1987) and with structural transitions alternating between inward- and outward-oriented states (Albers, 1967; Post et al, 1969). Ca²⁺-ATPases encompass the P2A and P2B subtypes of the P-type family (Palmgren and Nissen, 2011), consisting of typically ten membrane-spanning helical segments, M1-10, connected to three cytosolic domains: the nucleotide binding (N), the phosphorylation (P) with a catalytic aspartic acid residue conserved in all P-type ATPases, and the actuator (A) domain, which is involved in dephosphorylation and contains an essential TGES-motif containing loop (Anthonisen et al, 2006).

The mammalian sarco/endoplasmic reticulum Ca²⁺-ATPase (SERCA) is highly studied and often serves as a general model (Dyla et al, 2019a). SERCA explores an inward-open (E1) and an outward-open (E2P) conformation, exposing the ion-binding sites in the middle of the TM domain to either side of the membrane (Albers, 1967; Jardetzky, 1966; Lauger, 1979). In the E1 state, cytosolic calcium ions enter the binding site of SERCA via an entry pathway lined by M1-2, M4 and M6. Following Ca²⁺ binding at two cooperative sites, reorientation of the ATP-bound N domain facilitates transfer of the γ-phosphate from ATP to the catalytic aspartate in the P domain. Along with the phosphorylation of SERCA forming the [Ca₂]E1P state, the ion pathway closes and bound calcium becomes occluded (Sorensen et al, 2004; Toyoshima and Mizutani, 2004). Next, the A domain makes a large rotational movement perpendicular to the membrane, and the N domain moves away from the P domain in the formation of the outward-open E2P state. Altogether, these domain movements trigger the opening of the calcium exit pathway formed by M1-6. With calcium released to the other side of the membrane, protons occupy

[1]Department of Molecular Biology and Genetics, Aarhus University, Aarhus, Denmark. [2]The Danish Research Institute for Translational Neuroscience (DANDRITE), Nordic EMBL Partnership for Molecular Medicine, Aarhus, Denmark. [3]The Danish National Research Foundation Center for Proteins in Memory (PROMEMO), Aarhus, Denmark.
✉E-mail: pn@mbg.au.dk

the ion-binding sites and stimulate occlusion and dephosphorylation of the E2P state (Olesen et al, 2007), where the conserved TGES loop from the A domain moves in to catalyze dephosphorylation of the aspartyl phosphorylation site. Dephosphorylation and release of free phosphate allow a subsequent E2-E1 transition with the E1 state releasing counter-transported protons to the cytoplasmic side and opening again the cytoplasmic entry pathway (Winther et al, 2013).

However, structural information does not account for key steps and intermediates in the [Ca₂]E1P-ADP to E2P transition. When and how are ADP and calcium ions released, in which order, and how is an essentially irreversible step introduced in this process?

In SERCA, a four-glycine insert in the A-M1 linker (G₄-SERCA) allows occlusion of calcium in a stalled [Ca₂]E2P intermediate state (Daiho et al, 2010; Daiho et al, 2007). A similar intermediate was characterized by single-molecule Förster Resonance Energy Transfer (smFRET) data for the equivalent G₄ construct of the bacterial homolog Ca²⁺-ATPase 1 from *Listeria monocytogenes* (G₄-LMCA1), where smFRET traces indicated a calcium-bound, but ADP-insensitive [Ca]E2P state with a partially rotated A domain (Dyla et al, 2017). Subsequently, the A domain completes the rotation in the formation of the calcium-released E2P, which once reached showed no signs of reversal to the [Ca]E2P state. Hence, calcium release from the [Ca]E2P intermediate state was described as the critical, irreversible step. It points also to the A-M1 linker as crucial for transferring the energy from ATP hydrolysis to the extracellular release of calcium, as was also observed for SERCA (Daiho et al, 2010; Daiho et al, 2007). Other recent studies have described intermediate states for SERCA through time-resolved X-ray scattering (TR-XSS) (Ravishankar et al, 2020) and molecular dynamics (MD) (Thirman et al, 2021), generally confirming the rotation of the A domain as an important reaction coordinate for the E1P-E2P transition.

LMCA1 shares 34–39% sequence identity with mammalian SERCA (ATP2A genes), but it transports only a single calcium ion (Faxen et al, 2011) for each transport cycle (Fig. 1A), similar to the plasma membrane Ca²⁺-ATPase (PMCA, ATP2B genes) (Gong et al, 2018) and the secretory pathway Ca²⁺-ATPase (SPCA, ATP2C genes) (Chen et al, 2023; Dode et al, 2005) that share however a slightly lower sequence identity with LMCA1 (30-34%). We have previously reported crystal structures of LMCA1 stalled in presumably proton-occluded [H]E2-BeFₓ and [H]E2-AlFₓ forms and also G₄-LMCA1 in the [H]E2-BeFₓ form (Hansen et al, 2021). These structures were associated with a closed extracellular pathway and a TGES motif positioned for dephosphorylation (hence presumed all to be proton-occluded forms), which is unlike the SERCA E2-BeFₓ form with an outward-open calcium exit pathway. The LMCA1 structures were consistent with sequence alignments (Hansen et al, 2021) to indicate a single calcium-binding site (protonated, not calcium-bound) equivalent of the calcium-binding site II of SERCA, whereas the region corresponding to SERCA site I is occupied by Arg795 in LMCA1.

Here, we present a cryo-EM structure at 3.5 Å resolution for G₄-LMCA1 obtained under ATP turnover conditions and showing bound calcium at the predicted site. The intermediate represents a new structure for Ca²⁺-ATPases and ion-transporting P2-type ATPases in general, and is overall a hybrid between E1P and E2P states and a presumed intermediate in the E1P-E2P transition preceding the irreversible state of ion release.

# Results and discussion

LMCA1 was reconstituted in salipro nanodiscs (Frauenfeld et al, 2016), which serves both to embed the protein in a lipid environment and remove the detergent background signal in EM. Wild-type (WT) LMCA1 remains active when reconstituted in salipro nanodiscs as assessed by a phosphate-release assay (Fig. EV1), albeit with a slower rate compared to the detergent-solubilized form. In detergent, G₄-LMCA1 shows lower activity compared to WT (Dyla et al, 2017), but surprisingly, no Pᵢ release activity could be measured for saposin-solubilized G₄-LMCA1 (Fig. EV1). This may indicate that the saposin nanodisc stabilizes a particular state and/or restricts certain conformational changes occurring in the ATPase reaction for LMCA1 and in particular associated with the G₄ construct. Knowing these preconditions, we proceeded with a structural analysis of the sample.

## Overall conformation of cryo-EM structure

The published smFRET data for detergent-solubilized G₄-LMCA1 supplemented with 1 mM Ca²⁺ and 1 mM ATP revealed a predominant cycling between an E1-ATP and [Ca]E2P state showing high and medium-low FRET efficiency, respectively (Dyla et al, 2017). To investigate G₄-LMCA1 by cryo-EM, we incubated saposin-solubilized G₄-LMCA1 with 1 mM Ca²⁺ and 1 mM ATP at room temperature for 2–10 min before plunge-freezing and cryo-EM imaging. Expecting two or more conformations from the cryo-EM dataset, both 3D classification and 3D variability analyses were applied in a search for several conformations of G₄-LMCA1. However, only a single conformation was found after thorough processing of the data (Fig. 1B) and it showed clear indications of calcium binding and a hybrid conformation between E1P and E2P conformations (Fig. 1C–E). At the time of this study, no structures had been reported for LMCA1 in E1 states, so we constructed a homology model based on the [Ca₂]E1-AlFₓ-ADP structure of SERCA (pdb: 1T5T) (Data ref: Sorensen et al, 2004) for domain configurations. Previously, we published the crystal structure of the E2P state of LMCA1 based on an [H]E2-BeFₓ form (pdb: 6ZHH) (Data ref: Hansen et al, 2021). Since the E2P states of LMCA1 are different from SERCA and crucial for our analysis, and domain structures are quite different between LMCA1 and SERCA, we used the homology model of LMAC1 instead of the [Ca₂]E1-AlFₓ-ADP structure of SERCA to visually present the conformational changes from [Ca]E1P-ADP to E2P within LMCA1. During publication of our study, the Andersson laboratory published a cryo-EM structure of LMCA1 in a [Ca]E1-AlFₓ-ADP form, and it is consistent with the SERCA model at the level of detail required for the analysis presented (Prabudiansyah et al, 2024).

The structure of G₄-LMCA1 obtained here represents a new conformation (from now on termed [Ca]E2P) that overall is distinct from both the [Ca₂]E1-AlFₓ-ADP form of SERCA1a and a calcium-free E2P state of LMCA1 (RMSD values overall of 7.6 Å and 4.2 Å, respectively, Table EV1). Still, a RMSD of 2.9 Å for the cytoplasmic domains comparing the G₄-LMCA1 [Ca]E2P and the [H]E2-BeFₓ form of LMCA1 indicates a similarity of the cytoplasmic headpiece. Similarly, a RMSD value of 2.1 Å between the TM domain of the G₄-LMCA1 [Ca]E2P structure and SERCA1a [Ca₂]E1-AlFₓ-ADP (Sorensen et al, 2004) indicates a close similarity of the calcium-bound TM domain. Further visual

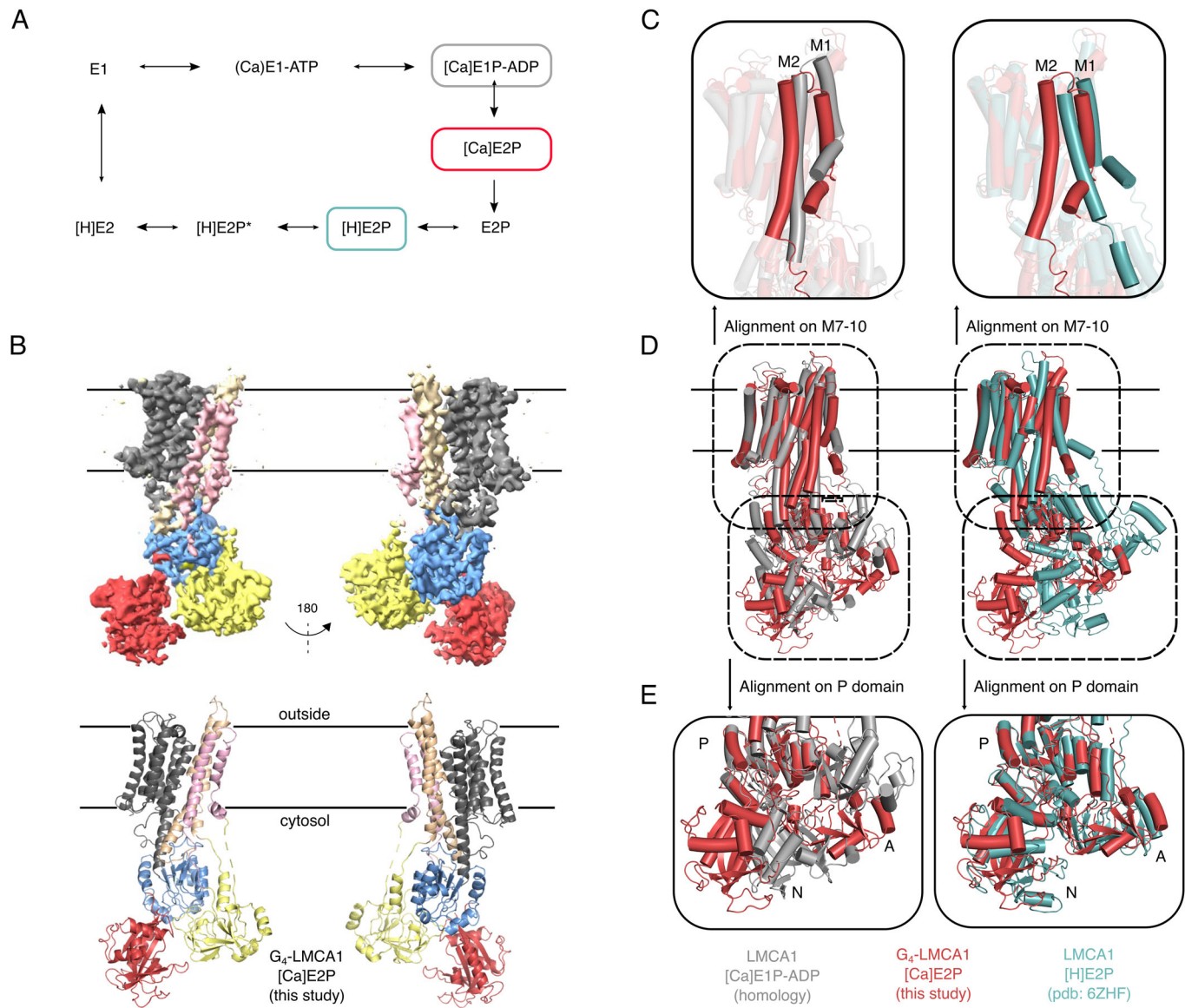

**Figure 1. The cryo-EM structure of G₄-LMCA1 adopts an intermediate E1P–E2P conformation.**

(A) Functional transport cycle of LMCA1. (B) 3.5 Å cryo-EM map and model of G₄-LMCA1 in the [Ca]E2P state. The map and model are color-coded according to the domains (N red, A yellow, P blue, M1-2 lightpink, M3-4 wheat, M5-10 gray). (C–E) Left panel: Alignment of [Ca]E2P (red) with an [Ca]E1P-ADP LMCA1 homology model based on SERCA [Ca₂]E1-AlFx-ADP crystal struture (pdb: 1T5T) (Data ref: Sorensen et al, 2004) (gray). Right panel: Alignment of [Ca]E2P (red) with LMCA1 [H]E2-BeFx crystal structure (pdb: 6ZHF) (Data ref: Hansen et al, 2021) (teal). The structures are aligned on M7-10 in (C–D) and on the P domain in (E). (C) Zoom-in on the TM domain. The TM of the [Ca]E2P state is positioned in an E1P-like conformation with M1-2 in a similar conformation as the [Ca]E1P-ADP state. (D) Overall conformation. (E) Zoom-in on the cytosolic domains. The cytosolic domains of [Ca]E2P is positioned in a similar conformation as the E2P state. The volume is made in ChimeraX (Pettersen et al, 2021) and models are made in PyMOL.

inspection reveals closed ion pathways, and density consistent with a single calcium ion bound and occluded at a binding site between M4 and M6 (Fig. 2B), equivalent of SERCA site II. Hence, the TM domain appears to adopt a calcium-occluded E1P-like conforma-tion, where M1-2 are not rotated into the E2P conformation yet (Fig. 1C).

Indeed, the cytosolic domains are not yet tilted relative to the membrane plane although configured in an E2P-like conformation with a much shorter distance between the TGES loop and the phosphorylation site when compared to the E1P conformation, as otherwise typical of the

E2P state (Fig. 1E). Phosphorylation of the catalytic aspartic acid residue in the P domain (Asp334 in LMCA1) is clearly visible (Fig. EV3), and the phosphorylation site is rotated away from the nucleotide-binding site of the N domain. Visual inspection of the cryo-EM density map indicates no density for bound nucleotide at the N domain despite a 1 mM background of ATP/ADP in the cryo-EM sample. This indicates that the N domain adopts an ADP-insensitive conformation in progression of a forward transport cycle, a hallmark of E2P states. Altogether, these observations point to the G₄-LMCA1 adopting an intermediate state of the transition between the inward-occluded [Ca]E1P-ADP state and the outward-open

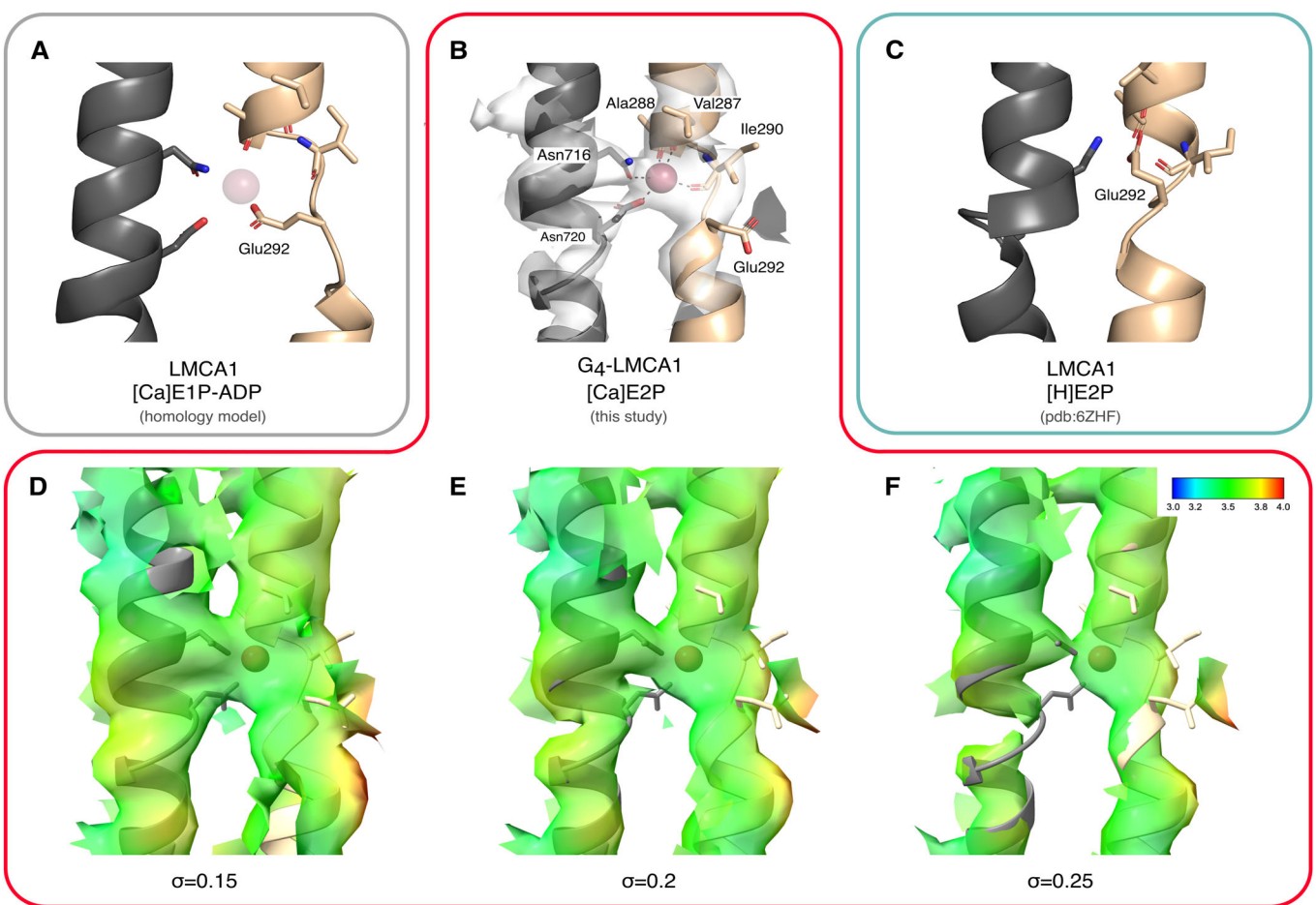

**Figure 2.  Glu292 coordination varies in different states.**

The calcium-binding site between M4 and M6 is shown for (A) the [Ca]E1P-ADP LMCA1 homology model based on SERCA [Ca$_2$]E1-AlF$_x$-ADP crystal structure (pdb: 1T5T) (Data ref: Sorensen et al, 2004), (B) the [Ca]E2P structure, and (C) the LMCA1 [H]E2P-BeF$_x$ crystal structure (pdb: 6ZHF) (Data ref: Hansen et al, 2021). (B) The electron density map is shown for M4, M6, and the calcium ion at contour level 10. Calcium-coordinating residues are shown as sticks. (D–F) Local resolution map of M4, M6, and the calcium ion at (D) σ = 0.15, (E) σ = 0.2 and (F) σ = 0.25.

E2P state, where ADP has been released and calcium is still bound. We denote this state [Ca]E2P, as it displays features of both [Ca]E1P and E2P, with the TM domain adopting a calcium-bound conformation and the cytoplasmic domains assuming an E2P-like, ADP insensitive conformation.

## The calcium ion is bound at a flexible site

As mentioned above, the structure of G$_4$-LMCA1 in the [Ca]E2P state is consistent with the suggested calcium-binding site equivalent of SERCA site II, i.e., between M4 and M6 (Fig. 2B). The calcium ion appears to coordinate two side chains in M6 (Asn716 and Asn720) and backbone carbonyl groups in M4 (Val287, Ala288, Ile290). Map features suggest that Asn716 is the primary calcium-coordinating side chain (Fig. 2D–F). Water molecules are likely also part of the coordination, although they could not be resolved in the 3.5 Å resolution cryo-EM map. For the [Ca]E1P-ADP state, Glu292 would be expected to point towards the calcium ion like for SERCA1a (Fig. 2A). However, the map for the [Ca]E2P form of G$_4$-LMCA1 reveals no density for the Glu292 side chain oriented towards the

calcium ion (Fig. 2B). Rather, Glu292 appears flexible and in a backbone conformation that directs the side chain away from the site. For comparison, earlier determined structures of LMCA1 with beryllium- or aluminiumfluoride represent subsequent calcium-released forms and show Glu292 now occupying the vacated calcium-binding site. Prediction of a high pK$_a$ for this side chain in the calcium-released forms indicates that it is most likely protonated, and that these forms represent proton-occluded [H]E2P-like states (Hansen et al, 2021) (Fig. 2C).

## Conformational changes for the [Ca]E1P-ADP to E2P transition

ATP hydrolysis overall drives the conformational changes that lead to calcium transport, and we will focus here on the transitions following the phosphorylation step, in particular the [Ca]E2P to E2P transition of LMCA1, which from smFRET data appears as the critical, irreversible step of calcium transport.

In the [Ca]E2P form of G$_4$-LMCA1, the P domain is placed at an intermediate position relative to the TM domain (Fig. 3E), and

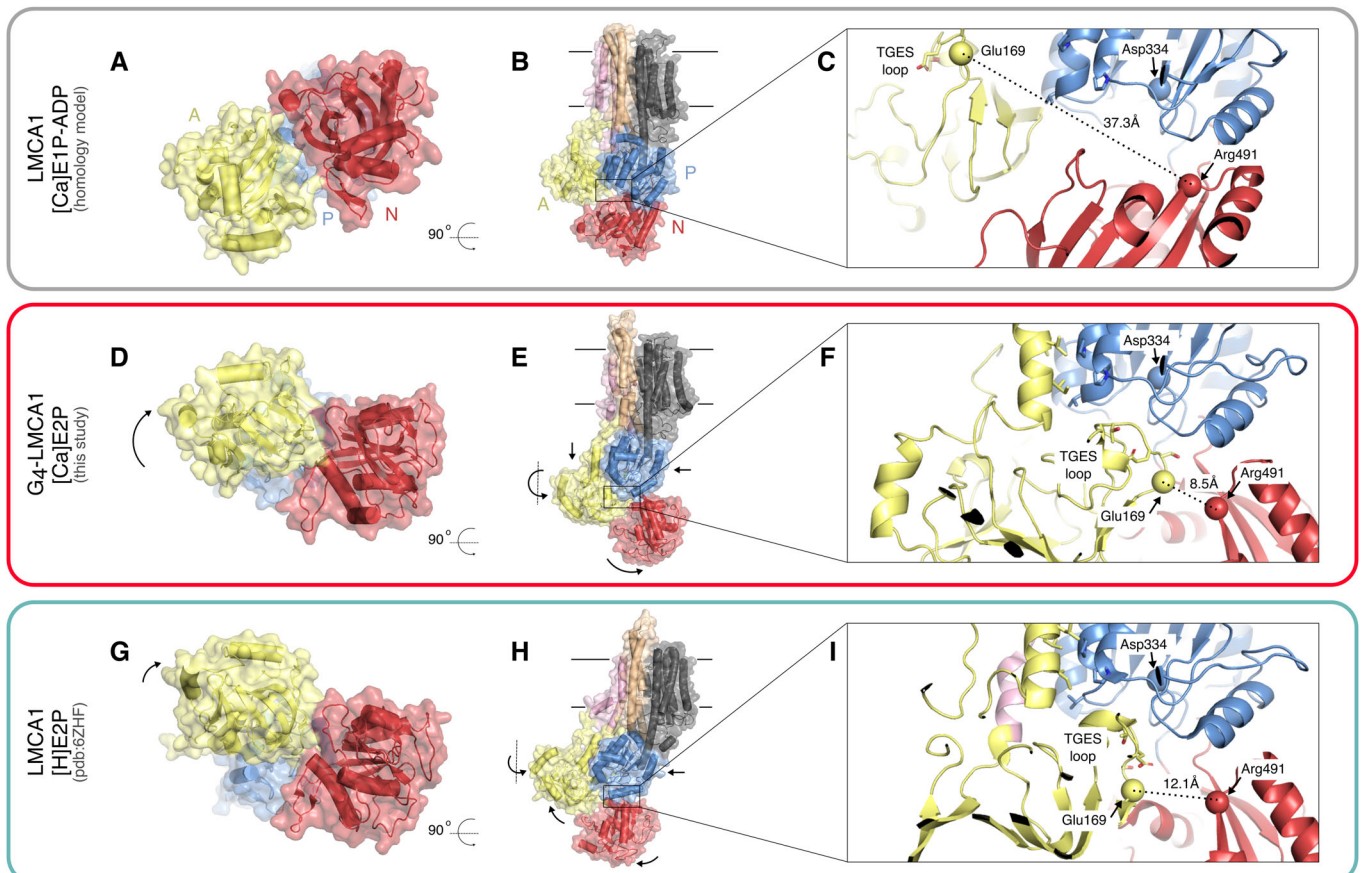

**Figure 3. Principal movements of cytosolic domains of LMCA1 during the E1P–E2P transition.**

Surface and cartoon representation of LMCA1 (**A–C**) [Ca]E1P-ADP LMCA1 homology model based on SERCA [Ca$_2$]E1-AlF$_x$-ADP crystal structure (pdb: 1T5T) (Data ref: Sorensen et al, 2004), (**D–F**) the G$_4$-LMCA1 [Ca]E2P structure, and (**G–I**) the LMCA1 [H]E2-BeF$_x$ crystal structure (pdb: 6ZHF) (Data ref: Hansen et al, 2021). The structures are colored according to Fig. 1B. Arrows indicate forward domain movements. (**A, D, G**) The structures represent the rotation of the A domain relative to the P domain. (**B, E, H**) The structures represent the movement of the cytosolic domains. Zoom-in panels show the TGES loop away from the phosphorylation site and a large distance between C$_\alpha$Glu169 and C$_\alpha$Arg491 (**C**), the TGES loop protecting the phosphorylation site and C$_\alpha$Glu169- C$_\alpha$Arg491 interaction (**F**), and the TGES loop primed for dephosphorylation and a break in the C$_\alpha$Glu169- C$_\alpha$Arg491 interaction (**I**). C$_\alpha$ of Asp334, Glu169 and Arg491 are shown as spheres. Residues in TGES loop and hydrophobic interactions between the A and P domain (Ile218, Leu221, Leu222 in the A domain and Pro601, Val625, Pro629 in the P domain) are shown as sticks in (**F, I**).

the A and N domain point away from the membrane, which puts G$_4$-LMCA1 in an upright conformation compared to [Ca]E1P-ADP (Fig. 3B) and [H]E2P (Fig. 3H). The A domain is partially rotated towards the position of the E2P state (Fig. 3A,D,G). The nucleotide-binding site is empty indicating that ADP must have been released prior to this point, probably at a preceding step going from the [Ca]E1P-ADP state to an ADP-sensitive [Ca]E1P state (Zhang et al, 2022).

The A domain in the [Ca]E2P intermediate of G$_4$-LMCA1 is rotated relative to the [Ca]E1-ADP conformation (Fig. 3C), and it now interacts with the P domain through hydrophobic interactions between the two domains (Fig. 3F,I). A salt bridge between the A domain (Glu169) and the N domain (Arg491) stabilizes the A domain in its intermediate position (Fig. 3F). When the A domain rotates further into the E2P state, this interdomain bond is disrupted, but the hydrophobic interactions between the A and P domain remain (Fig. 3I). In the [Ca]E2P state, the TGES loop in the A domain blocks access to the phosphorylated aspartate in the P domain for a reverse reaction with ADP (Fig. 3F). However, unlike

the calcium-released [H]E2P state, where the pump is primed for dephosphorylation (Hansen et al, 2021), the TGES motif of [Ca]E2P is not yet positioned for catalysis at the phosphorylated aspartic acid residue; rather phosphorylation is shielded (Fig. 3I).

## The A-M linkers couple cytosolic domains to TM movement

Energy deposited by ATP-mediated phosphorylation must be transmitted during the [Ca]E1P-ADP to E2P transition. Both biochemical and computational studies suggest that energy is stored in the linkers between the A domain and the transmembrane helices M1 and M2 (and M3) and that the strain built up transmits to the subsequent opening of the TM domain and release of calcium (Daiho et al, 2007; Thirman et al, 2021).

Figure 4 highlights the variation of the length spanned and the conformation of the A-M1 and M2-A linkers. The A-M1 linker is unstructured without any apparent interactions in [Ca]E1P-ADP (model based on SERCa1a crystal structures) and in the crystal

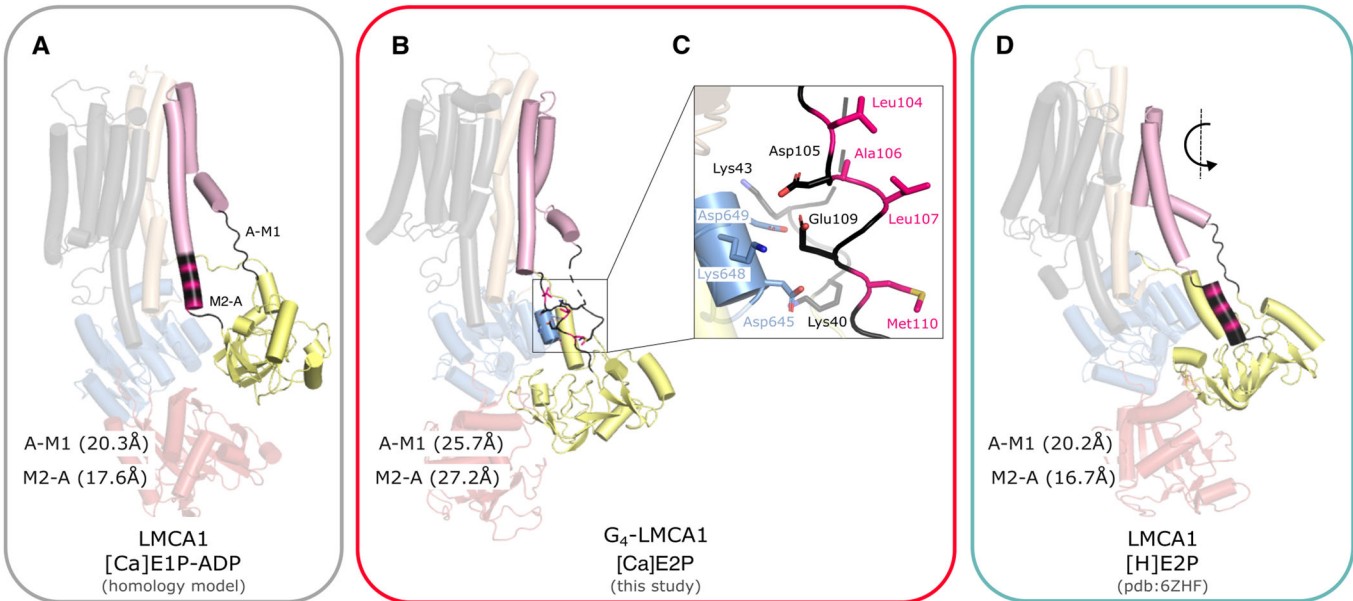

**Figure 4. A-M2 linker changes its conformation during the E1P–E2P transition.**

(A) [Ca]E1P-ADP LMCA1 homology model based on SERCA [Ca$_2$]E1-AlF$_x$-ADP crystal structure (pdb: 1T5T) (Data ref: Sorensen et al, 2004), (B) the G$_4$-LMCA1 [Ca]E2P structure, (C) Zoom-in panel showing possible ionic interactions between A-M1, M2-A, and residues in the P domain, and (D) the LMCA1 [H]E2-BeF$_x$ crystal structure (pdb: 6ZHF) (Data ref: Hansen et al, 2021). The structures are colored according to Fig. 1B. The A-M1 linker (Leu39-Pro46) and the M2-A linker (Ser102-Pro113) are colored black, and the distance between C$_\alpha$Leu39- C$_\alpha$Pro46 and C$_\alpha$Ser102- C$_\alpha$Pro113, respectively, is shown. The four hydrophobic residues (Leu104, Ala106, Leu107, and Met110) are clored pink. Arrow indicate forward domain movements.

structure of [H]E2-BeF$_x$ representing a [H]E2P state of LMCA1. The same is true for the [Ca]E2P intermediate state with an A-M1 linker that is extended by four glycines. However, the distance spanned by this linker is almost 6 Å longer than in the [Ca]E1P-ADP and E2P states (Fig. 4A,B,D). The WT linker lacking the G$_4$ insert cannot span this gap without a conformational rearrangement. To assess if the WT linker can in principle link the configuration of domains in the [Ca]E2P intermediate state, we allowed it to deviate locally from the G$_4$-insert form (Fig. EV4). This does not necessarily capture the actual structure of the linker and M1 in the WT [Ca]E2P state, but it shows that the WT protein can in principle assume such an intermediate conformation in a physically realistic form. Likely it represents a strained, high-energy state.

Furthermore, four hydrophobic residues (Leu104, Ala106, Leu107, Met110) positioned in a helical part of the M2-A linker interact in a hydrophobic network with M4 and the P domain in the modeled [Ca]E1P-ADP state of LMCA1 (based on SERCA1a [Ca$_2$]E1-AlF$_x$-ADP, Fig. 4A). This network is disrupted in the [Ca] E2P structure, where the helix is unwound and stretched by almost 10 Å. The change allows the A domain to obtain a partially rotated position in [Ca]E2P compared to the fully rotated [H]E2P state (Fig. 4B). The four hydrophobic residues in the M2-A linker twist away from the P domain interface, and a network of ionic interactions is formed between M2-A and the P domain (Fig. 4C) stabilizing the M2-A linker in the extended conformation. In the calcium-released [H]E2P state, M1-M2 are rotated and the M2-A linker is again shortened by helix formation, which incorporates the four hydrophobic residues that formed a hydrophobic network with the A domain in the [Ca]E2P intermediate (Fig. 4D).

## A transient interaction in the E1–E2 transition

Several intermediates for LMCA1 are explored that couple ATP hydrolysis to calcium release (Fig. 5). Upon phosphoryl transfer from ATP and then ADP release, the cytosolic domains can reorientate from E1P into an E2P-like configuration leading to the [Ca]E2P intermediate presented here. The TGES loop in the A domain now shields the phosphorylation site in the P domain, and the empty nucleotide-binding site in the N domain and the A–N interdomain interactions are stabilized by the ionic interation between Glu169 of the A domain and Arg491 of the P domain (Fig. 3F). Arg491 in LMCA1 is conserved in all P2-type ATPases and corresponds to Arg560 in SERCA1a, which is important for nucleotide binding (Espinoza-Fonseca and Thomas, 2011; Sorensen et al, 2004; Toyoshima and Mizutani, 2004). A SERCA1a Arg560Ala mutation is known to stimulate the transition to the calcium-free E2P state (Clausen et al, 2003), which could be explained by the loss of the interaction that stabilizes the preceding state. This suggests that Arg491 in the N domain has a dual role of nucleotide binding in the [Ca]E1-ATP and [Ca]E1P-ADP states, and thereafter of A domain stabilization in the transient [Ca]E2P state that prepares for Ca$^{2+}$ release.

As mentioned earlier, the Glu292 residue of the calcium site appears flexible in the [Ca]E2P state (indicated by poor density and an orientation away from the Ca$^{2+}$ site), and hence the calcium site has lower coordination by protein side chains, which may prime solvation of the Ca$^{2+}$ ion and its release in the subsequent step. At the same time, the flexible Glu292 side chain can pick up a proton, presumably from the extracellular environment, and enter the vacant site in a neutral form after Ca$^{2+}$ release, i.e., leading to

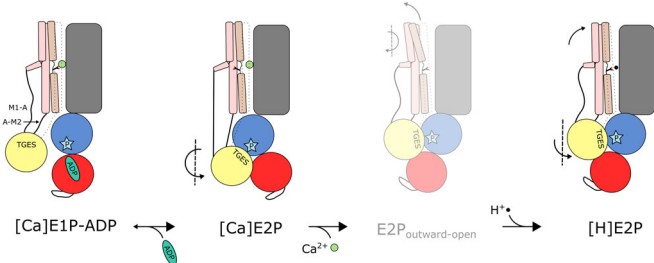

**Figure 5. Sequence of events in the E1–E2 transition.**

Schematic representation of the intermediate conformations during the E1–E2 transition. The E2P$_{outward-open}$ conformation is in faint colors to indicate that it is transient for LMCA1. The representations are color-coded according to the domains (N red, A yellow, P blue, M1-2 light pink, M4 wheat, M5-10 gray). M3 and M3-A are only indicated with dotted lines for clarity. Glu292 on M4 is shown in a stick representation.

proton occlusion and dephosphorylation. Glu292 corresponds to Glu309 in SERCA, which is also conserved in all P2-type ATPases; essential for calcium binding and gating (Andersen and Vilsen, 1992; Clausen et al, 2013; Inesi et al, 2004).

Contrary to wild-type ATPases, the four-glycine insert in the A-M1 linker relaxes the [Ca]E2P state of G$_4$-LMCA1, slowing down its transition to release calcium. Therefore, it accumulates in the [Ca]E2P state as revealed here by cryo-EM under ATPase turnover conditions in a lipid nanodisc. At the same time, the nanodisc appears to block the ATPases activity of G$_4$-LMCA1 (which shows an impaired, but still significant activity in detergent) by stabilizing this particular state, although particular mechanisms for this cannot be detailed here due to a very low resolution of the map for saposin nanodisc features. The A-M1 linker plays a crucial role in the formation of a calcium exit pathway, which is in agreement with already published experiments on phosphorylation and calcium release kinetics (Daiho et al, 2010; Daiho et al, 2007), smFRET (Dyla et al, 2017) and molecular dynamics (Thirman et al, 2021). Figure EV4 shows the density for the A-M linkers of G$_4$-LMCA1 in the [Ca]E2P structure. Parts of the M2-A linker are clearly defined in density and dock to the P domain, and these interactions likely stabilize the [Ca]E2P state too. Presumably, both the A-M1 and the M2-A linker stretch in the wild-type [Ca]E2P intermediate and as a next, they "pull open" a calcium exit pathway by strain relaxation. For SERCA, this open E2P state forms a stable intermediate, whereas for LMCA1 it will immediately close with proton occlusion. Unlike for SERCA, calcium release in LMCA1 therefore is associated with direct rotation of the A domain in the formation of a closed, proton-occluded [H]E2P state, with a proton likely picked up at Glu292 in a direct exchange for Ca$^{2+}$, as also indicated earlier by crystal structures (Hansen et al, 2021). SERCA, in contrast, likely features a back-door pathway for proton entry (Bublitz et al, 2013) and explores an outward-open E2P state prior to proton occlusion at two sites that may even co-operate.

## Calcium release is the irreversible step for LMCA1

Earlier smFRET studies (Dyla et al, 2017) and the studies presented here indicate that the LMCA1 [Ca]E2P state is the critical

intermediate that precedes the practically irreversible transition of the LMCA1 transport cycle. The A domain completes its rotation only when calcium is released (Dyla et al, 2017), and presumably it is linked to very fast kinetics of proton occlusion, as described above for Glu292, and the subsequent dephosphorylation reaction of LMCA1 once the [H]E2P state is reached. Indeed, our crystal structure of the [H]E2-BeF$_x$ state shows that the enzyme is primed for dephosphorylation with the glutamate side chain of the TGES already positioned over the phosphorylation site to catalyze the in-line attack of a water molecule (Hansen et al, 2021). The G$_4$-LMCA1 [H]E2-BeF$_x$ form does not enter a [Ca]E2P-like state in reverse reaction with Ca$^{2+}$ present (Dyla et al, 2017; Hansen et al, 2021), unlike what SERCA has been reported to do (Daiho et al, 2010; Daiho et al, 2007). The outward-open E2P state therefore is short-lived for LMCA1, and the kinetics are too fast to determine a single irreversible step. Hence, the most likely irreversible transition for LMCA1 is [Ca]E2P to [H]E2P, which completes the A domain rotation transmitted by linker regions, and which includes calcium release with no outward-open Ca$^{2+}$ intermediate explored per se.

## Comparing LMCA1 and SERCA

We have determined a structure of the so far elusive calcium-occluded and ADP-insensitive [Ca]E2P intermediate for LMCA1 by cryo-EM using a G$_4$ insertion mutant form, which is known to pause at this intermediate. The TM domain maintains an E1P-like conformation with bound calcium—for LMCA1, a single calcium ion at the site corresponding to site II in SERCA. The cytosolic domains adopt an E2P-like configuration with a partially rotated A domain, and the ADP nucleotide is released. However, the cytoplasmic headpiece of the three cytoplasmic domains is still not tilted into the configuration of the calcium-released E2P state, but rather paused in the intermediate state. The pause is likey caused by the G$_4$-extension of the A-M1 linker relaxing strain. G$_4$-SERCA is prone to reverse reactions with BeF$_x$ and Ca$^{2+}$ to form a [Ca$_2$]E2-BeF$_x$ form (Daiho et al, 2010), and even P$_i$ and Ca$^{2+}$ can reverse it into the [Ca$_2$]E2P state (Daiho et al, 2007). This indicates that the transition from [Ca$_2$]E2P to E2P is not an irreversible step for SERCA. The related P2C-type Na$^+$,K$^+$-ATPase in the E2-BeF$_x$ form can also reverse into sodium-occluded E2P/E1P-like states, stimulated by ADP (Fruergaard et al, 2022). In SERCA, the outward-open E2-BeF$_x$ structure reveals that the A domain is not yet fully rotated (Olesen et al, 2007). This suggests that when the ion-binding sites of SERCA remain accessible in this outward-open state (exposed to the controlled environment of the sarco/endoplasmatic lumen), the nucleotide-binding site might still require protection from a reverse ADP reaction, provided by the TGES loop in the A domain.

Probably, the irreversible step of the SERCA cycle appears at a next step of proton-occluding closure of the extracellular pathway, where the A domain completes its rotation, dephosphorylation takes place, and the cycle proceeds with P$_i$ release and modulatory ATP (Jensen et al, 2006). The final rotation coupled with proton occlusion allows the glutamate in the TGES loop to dephosphorylate the phosphorylation site. Molecular dynamics (MD) simulations and single-molecule studies could be informative approaches for future studies into these critical partial reactions.

The minimal exposure of an outward-open state for bacterial LMCA1 may be an important mechanism to avoid reversible function or damage in a potentially harsh and uncontrolled extracellular environment, and it may also prevent any $Ca^{2+}$-ATPase inhibitors from the surrounding environment to explore binding sites in the outward-open pathway. It may therefore have been selected for in bacterial evolution. The LMCA1 studies presented and discussed here add valuable information on both P-type ATPase mechanisms in general and $Ca^{2+}$-ATPases in bacteria specifically.

# Methods

### Reagents and tools table

| Reagent/resource | Reference or source | Identifier or catalog number |
| --- | --- | --- |
| **Experimental model** | | |
| C43 (DE3) cells (E. coli) | Biosearcg technologies | 60446-1 |
| **Recombinant DNA** | | |
| pET-22b | Novagen | 69744 |
| **Oligonucleotides and sequence-based reagents** | | |
| PCR primers | Sigma | |
| **Chemicals, enzymes, and other reagents** | | |
| Mutagenesis kit | Bionordika | E0554S |
| LB medium mix | Th. Geyer | 8891 |
| Ampicillin | Fisher | 10419313 (BP1760-25) |
| Tris | Vwr | 103156X |
| HEPES | Vwr | 441487M |
| MOPS | Fisher | 10234723 |
| NaCl | Vwr | 27810295 |
| KCl | Vwr | 26764298 |
| $MgCl_2$ | Vwr | 87060.290 |
| $CaCl_2$ | Vwr | 22328.262 |
| BME | Merck | M6250 |
| EGTA | Fisher | 409911000 |
| Imidazole | Fisher | 10561331 |
| Glycerol | Vwr | 24388295 |
| PMSF | Th. geyer | 6367-100G |
| DNaseI | Merck | 10104159001 |
| $C_{12}E_8$ | Anatrace | O330 |
| SDS | Merck | 75746 |
| Brain extract lipids | Merck | B1502 |
| DDM | Inalco | 1758-1350 |
| LMNG | Bionordika | ng310 |
| Saposin A | Lyons et al, 2017 | N/A |
| ATP | Fisher | 15470177 (J61125.14) |
| Ascorbic acid | Sigma | A7506 |

| Reagent/resource | Reference or source | Identifier or catalog number |
| --- | --- | --- |
| $(NH_4)_2MoO_4$ | Acros Organics | 205851000 |
| Sodium arsenite | Fisher | S/2330/48 |
| Sodium citrate | Millipore | 1.06448.0500 |
| Acetic acid | Fisher | 10304980 (A/0400/PB17) |
| Ni-sepharose 6 Fast Flow | GE Healthcare | 17-5318-06 |
| **Software** | | |
| Excel | Microsoft | |
| EPU | Thermo Scientific | |
| CryoSPARC | Punjani et al, 2017 | |
| Namdinator | Kidmose et al, 2019 | |
| Phenix | Liebschner et al, 2019 | |
| PyMOL | Schrödinger | |
| ChimeraX | Pettersen et al, 2021 | |
| Modeller | Sali and Blundell, 1993 | |
| MUSCLE | Edgar, 2004 | |
| Coot | Emsley and Cowtan, 2004 | |
| **Equipment** | | |
| Tabletop centrifuge | Eppendorf | |
| Heraeus mutlifuge X1R | Thermo Fisher | |
| Sorvall Ultracentrifuge WX+ | Thermo Scientific | |
| Optima MAX-XP Ultracentrifuge | Beckman | |
| TLA-110 rotor | Beckman | |
| T-647.5 rotor | Thermo Scientific | |
| ÄKTApurifier | Cytiva | |
| SDS-PAGE equipment | Biorad | |
| VICTOR3 plate reader | Perkin Elmer | |
| GloQube Glow discharger | Quorum Technologies | |
| EM GP2 plunge freezer | Leica | |
| Titan Krios G3i microscope | Thermo Fisher Scientific | |
| BioQuantum energy filter | Gatan | |
| K3 camera | Gatan | |
| **Others** | | |
| XK-16 gravity flow column | GE Healthcare | |
| Vivaspin MWCO 50 kDa | GE Healthcare | GE28-9323-62 |
| Gel | Merck | MP8W12 |
| Superdex 200 10/300 | GE Healthcare | 28-9909-44 |
| ELISA plate 96-wells | SARSTEDT AG & Co. KG | 82.1583.100 |
| CF-2/2-3Cu grids | Protochips | CF-222C |
| CF-1.2/1.3-3Cu grids | Protochips | CF313 |

G$_4$-LMCA1 was constructed, expressed, and purified using His-tag affinity and size-exclusion chromatography as described earlier (Hansen et al, 2021).

## Reconstitution in salipro nanodiscs

In total, 75 µl of detergent-solubilized G$_4$-LMCA1 at 10 mg/ml in buffer (50 mM Tris, 200 mM KCl, 10 mM MgCl$_2$, 20% v/v glycerol, 1 mM BME, 0.25 mg/mL C$_{12}$E$_8$, pH = 7.6) was mixed with 75 µl brain extract lipids at 5 mg/ml and incubated at room temperature for 30 min. The brain extract lipids were prepared at 20 mg/ml in buffer (1.5% w/v DDM, 50 mM Tris-HCl, 150 mM KCl, pH 7.5) and diluted four-fold to obtain a concentration of 5 mg/ml. In total, 250 µl of saposin A at 5 mg/ml in buffer (20 mM HEPES, 150 mM NaCl, pH 7.5) was added to the mix and incubated for 10 min at room temperature. Saposin A was expressed and purified according to earlier reports (Lyons et al, 2017). The reconstituted monomeric fraction was separated from aggregates and empty discs on a Superdex 200 10/300 in buffer (50 mM Tris-HCl, 150 mM KCl, pH = 7.6).

Reconstitution of WT LMCA1 used for ATPase activity assay was carried out similar to G$_4$-LMCA1, but worked best in small reaction volumes, so 17 parallel reactions with 5 µl protein, 5 µl brain extract lipids and 20 µl saposin A were mixed under the same conditions. After reconstitution, the small reactions were pooled, and the monomeric fractions were collected from a Superdex 200 10/300. The results are shown in Fig. EV1.

## ATPase activity assay

The ATPase activity was measured by the Baginski method (Baginski et al, 1967) with quantification of inorganic phosphate released by LMCA1 after reaction with CaCl$_2$ and ATP. The reaction mix was started by the addition of ATP (25 mM stock) to a 3 mM ATP final concentration in an Eppendorf tube. The reaction mixture consisted of 5 µg/ml LMCA1, 1.1 mM CaCl$_2$, 0.1 mM EGTA, 50 mM Tris, 200 mM KCl, 10 mM MgCl$_2$, 20% v/v glycerol, 1 mM BME, 0.25 mg/mL C$_{12}$E$_8$, pH = 7.6 and the sample reconstituted in nanodiscs was in a reaction mixture without detergent and glycerol (5 µg/ml LMCA1, 1.1 mM CaCl$_2$, 0.1 mM EGTA, 50 mM Tris, 200 mM KCl, 10 mM MgCl$_2$, 1 mM BME, pH = 7.6). The negative control contained 1.1 mM EGTA and no CaCl$_2$. After 2–8 min, 50 µl of the reaction mix was terminated in a 96-well plate with 50 µl freshly prepared stop solution (140 mM ascorbic acid, 5 mM (NH$_4$)$_2$MoO$_4$, 0.1% w/v SDS, 0.4 M HCl) in each well and incubated for 15 min. The blue color formed was stabilized with 75 µl arsenite solution (150 mM sodium arsenite, 70 mM sodium citrate, 350 mM acetic acid). After 30 min of incubation, the absorbance was measured at 860 nm and the activity was calculated.

## Cryo-EM grid preparation and data collection

The sample mix (0.7 mg/ml saposin-solubilized G$_4$-LMCA1, 50 mM Tris, 150 mM KCl, 1 mM CaCl$_2$, 10 mM MgCl$_2$, 0.0015% LMNG, pH = 7.6) was supplemented with 1 mM ATP and incubated at room temperature for 2–10 min. In all, 3 µl were added to a freshly glow-discharged grid (45 s at 15 mA), which was subsequently blotted at 20 °C and 90% humidity for 4 s. CF-2/2-3Cu (Protochips) were used, except for CF-1.2/1.3-3Cu in one case. Blotting and subsequent plunge-freezing into liquid ethane were carried out on an EM GP2 plunge freezer (Leica) with one or two filter papers.

Data were collected on a Titan Krios G3i microscope (EMBION Danish National cryo-EM Facility – Aarhus node) operated at 300 KeV equipped with a BioQuantum energy filter (energy slit width 20 eV) and K3 camera (Gatan). Movies were collected using aberration-free image shift data collection (AFIS). A nominal magnification of 130,000x was used, resulting in a pixel size of 0.647 Å$^2$/px with a total dose of 59.0 e$^-$/Å$^2$ (grid 1 + 2) and 58.5 e$^-$/Å$^2$ (grid 3 + 4). The movies were fractionated into 53 frames (1.13 e$^-$/Å$^2$ per frame) at a dose rate of ~18 e$^-$/px/s and a 1.4 s exposure time per movie. A nominal defocus range of $-0.8$ to $-2.0$ µm was used.

## Cryo-EM image processing

Figure EV2 shows the processing pipeline for image processing. Movies were motion corrected (Rubinstein and Brubaker, 2015) and CTF estimated (Zivanov et al, 2020) using CryoSPARC (v3 and v4) (Punjani et al, 2017). Particles from grid 1 were picked using a circular blob and aligned by 2D classification. Small subsets of the 2D classes were selected and used to generate ab initio volumes. One protein-like and one–two junk volumes were used to select protein particles from junk in multiple rounds of hetero-genous refinement. This particle stack was used as a template for template picking in all micrographs, which were merged from four grids to extract 3,001,118 particles. Again, particles from selected 2D classes were used to generate ab initio volumes for separating protein from junk particles in multiple rounds of heterogeneous refinement. Global CTF refinement followed by non-uniform refinement (Punjani et al, 2020) of 101,034 particles generated a volume that was used to manually generate a mask in ChimeraX (Pettersen et al, 2021). This mask was used in 3D variability (Punjani and Fleet, 2021) in cluster mode. One of the clusters provided a volume that was finally used for non-uniform refinement.

## Homology modeling

A homology model for the [Ca]E1P-ADP state of LMCA1 was made using an online version of Modeller (www.salilab.org/modeller) (Sali and Blundell, 1993). A sequence alignment of LMCA1 and SERCA performed in MUSCLE (Edgar, 2004) was used as an input file together with the structure of SERCA stabilized with Ca$^{2+}$, ADP, and AlF$_x$ (pdb: 1T5T) (Data ref: Sorensen et al, 2004). The model was used for comparative illustrations of domain configurations, and no detailed interactions e.g., at the nucleotide-binding site were modeled and considered.

## Model building, refinement, and validation

A fusion model was created by using the cytosolic domain from the crystal structure of G$_4$-LMCA1 E2P (pdb: 6ZHH) (Hansen et al, 2021) and the TM domain of the LMCA1 homology model for the [Ca$_2$]E1P-ADP state (pdb: 1T5T) (Data ref: Sorensen et al, 2004). The model was manually docked in the cryo-EM volume and fitted to the map with geometry restraints using Namdinator (Kidmose et al, 2019).

Real-space refinement of the structure was done in Phenix (Liebschner et al, 2019), and model building and analysis were performed in Coot (Emsley and Cowtan, 2004) (Table EV2). A DeepEMhancer (Sanchez-Garcia et al, 2021) map further guided model building.

## Data availability

The atomic coordinates for the structure have been deposited in the Protein Data Bank (PDB) under the accession code: 9GQO. The electron microscopy density map has been deposited in the Electron Microscopy Data Bank (EMDB) under the accession code: EMD-51510. These data can be accessed at the PDB (https://www.rcsb.org/structure/9GQO) and EMDB (https://www.ebi.ac.uk/emdb/EMD-51510) repositories, respectively.

The source data of this paper are collected in the following database record: biostudies:S-SCDT-10_1038-S44319-025-00392-x.

## Peer review information

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

## Acknowledgements

The authors dedicate this paper to the memory of the late professor Jesper Vuust Møller (1938-2023), who founded studies of $Ca^{2+}$-ATPase at Aarhus University in the 1970's and with whom the authors have had numerous fruitful discussions, collaborations and advice on $Ca^{2+}$-ATPases. The authors are grateful to Jens Peter Andersen for the critical reading of the manuscript. The authors would like to thank Anna Marie Nielsen for technical assistance, to Josephine Karlsen Dannersø and the staff of the EMBION cryo-EM facility for help with sample preparations, optimization and data collection, and to Samuel Hjort-Jensen and Jesper Karlsen for help with data processing and structure determination using the EMCC facilities at Aarhus University for scientific computing. This work was supported by a PhD scholarship from the Boehringer Ingelheim Fonds to SBH, a professorship grant from the Lundbeck Foundation (R310-2018-3713) to PN, and a project 2 grant from the Danish Fund for Independent Research (7014-00328B) to MK and PN. The cryo-EM research infrastructure was supported with grants from the Danish Ministry for Research and Higher Education (EMBION grant no. 5072-00025B), and the Novo Nordisk Foundation (ICE-T grant no. NNF20OC0060483).

## Author contributions

**Sara Basse Hansen**: Formal analysis; Funding acquisition; Investigation; Visualization; Writing—original draft; Writing—review and editing. **Rasmus Kock Flygaard**: Data curation; Formal analysis; Supervision; Methodology. **Magnus Kjærgaard**: Formal analysis; Supervision; Funding acquisition; Investigation; Methodology. **Poul Nissen**: Conceptualization; Resources; Formal analysis; Supervision; Funding acquisition; Project administration; Writing—review and editing.

Source data underlying figure panels in this paper may have individual authorship assigned. Where available, figure panel/source data authorship is listed in the following database record: biostudies:S-SCDT-10_1038-S44319-025-00392-x.

## Disclosure and competing interests statement

The authors declare no competing interests.

# Expanded View Figures

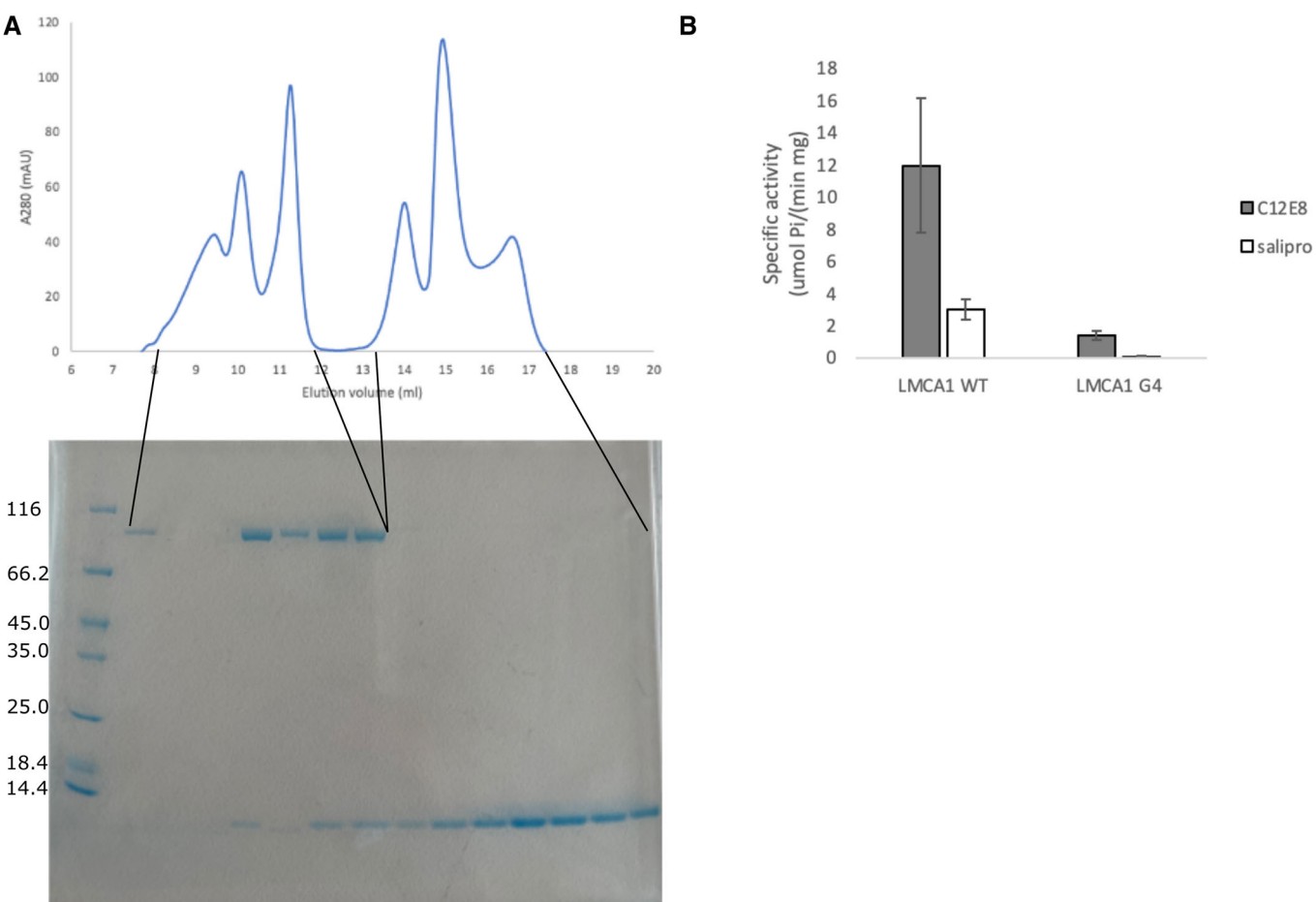

**Figure EV1. Salipro reconstitution and activity.**

(A) Chromatogram of salipro reconstitution of LMCA1. The peak eluting at 11–12 ml contains monomeric LMCA1 in nanodiscs. (B) Specific ATPase activity is obtained by linear regression of a time-course measurement. Error bars represent the standard deviation of three technical replicates.

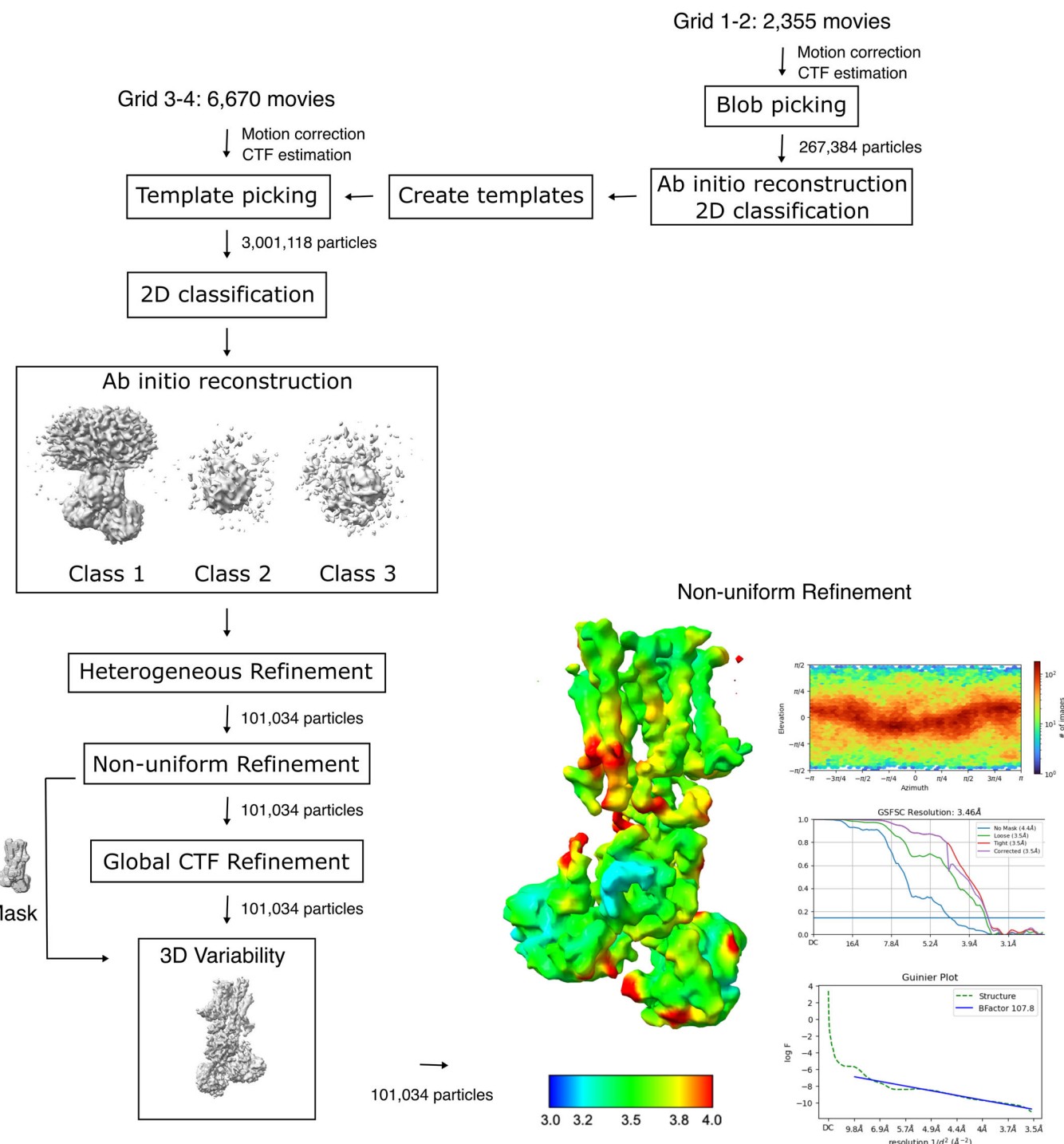

**Figure EV2.   The CryoSPARC data processing pipeline for cryo-EM single particle analysis.**

Data from four grids were merged into a single dataset and processeed. The final struture from a non-uniform reefinement is shown with its local resolution represented by a various colors. Plots from the final refinement are shown.

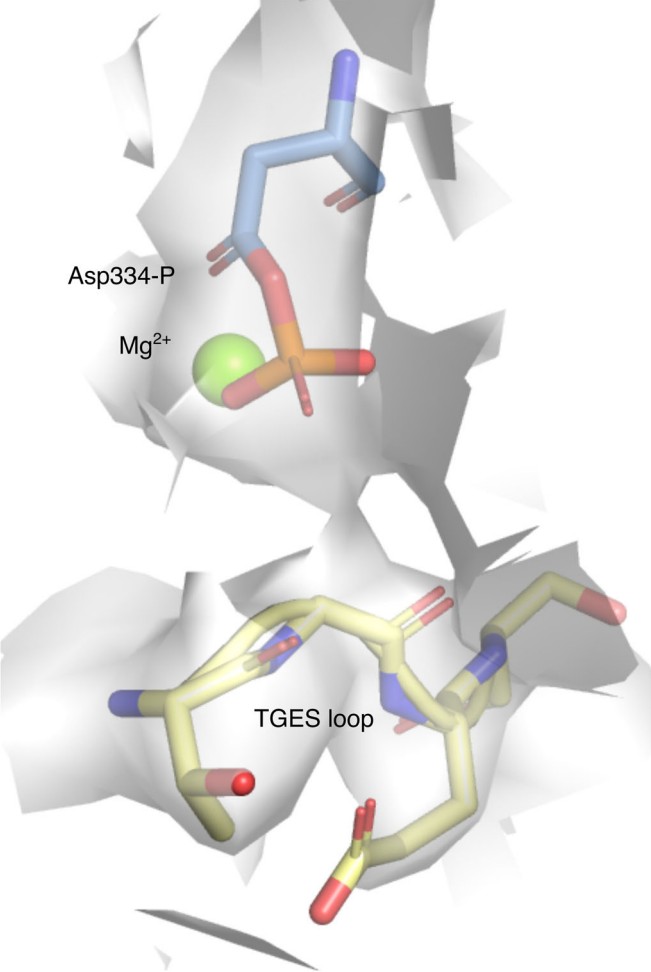

**Figure EV3.   Density for the phosphorylation site.**

The catalytic aspartate (Asp334) is phosphorylated and coordinated by a $Mg^{2+}$ ion. The TGES loop is protecting the phosphorylation site. Density is only shown for the TGES loop in yellow, Asp334 in blue, the phosphorylation and the $Mg^{2+}$ ion.

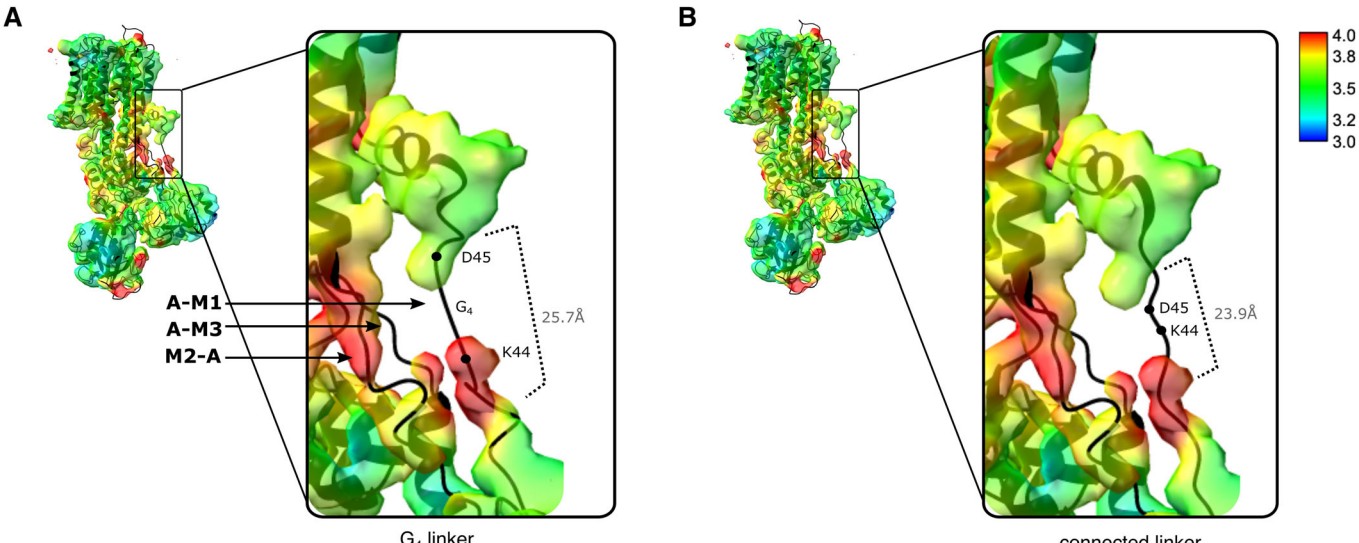

**Figure EV4.   The A-M1 linker can span the domains in a strained conformation in WT LMCA1 in [Ca]E2P.**

(A) The break in linker A-M1 is indicated with numbered residues, and the $C_\alpha$Leu39 - $C_\alpha$Pro46 distance is measured. (B) Lys44 and Asp45 can connect without the $G_4$ insert by remodeling the loop in coot. The linker can span the domains in an extended conformation if a local deviation from the density is allowed. The $C_\alpha$Leu39 - $C_\alpha$Pro46 distance is measured like in Fig. 4. The map is shown at $\sigma = 0.225$.

