## [Peer Review File · EMBO Reports]

Structure of the [Ca]E2P intermediate of Ca²⁺-ATPase 1 from *Listeria monocytogenes*

Poul Nissen, Sara Basse Hansen, Rasmus Flygaard, and Magnus Kjærsgaard

Corresponding author(s): Poul Nissen (pn@mbg.au.dk)

Review Timeline:

Submission Date:	22nd May 24
Editorial Decision:	5th Jul 24
Revision Received:	20th Oct 24
Editorial Decision:	19th Nov 24
Revision Received:	5th Jan 25
Accepted:	22nd Jan 25

Editor: Deniz Senyilmaz Tiebe

Transaction Report:

Dear Prof. Nissen,

Thank you for submitting your manuscript to EMBO Reports. My apologies for this unusual delay in getting back to you. Three referees agreed to review your manuscript. So far, we have received two referee reports that are copied below. Given that both referees are in fair agreement that you should be given a chance to revise the manuscript, I would like to ask you to begin revising your study along the lines suggested by the referees.

Please note that this is a preliminary decision made in the interest of time, and that it is subject to change should the third referee offer very strong and convincing reasons for this. As soon as we receive the final report on your manuscript, we will forward it to you as well.

Referees express interest in the presented [Ca]E2P intermediate of Ca²⁺-ATPase 1 from *Listeria monocytogenes*. However, the also raise some concerns that need to be addressed to consider publication here.

Given these positive recommendations, we would like to invite you to revise your manuscript with the understanding that the referee concerns (as in their reports) must be fully addressed and their suggestions taken on board. Please address all referee concerns in a complete point-by-point response. Acceptance of the manuscript will depend on a positive outcome of a second round of review. It is EMBO reports policy to allow a single round of major experimental revision only and acceptance or rejection of the manuscript will therefore depend on the completeness of your responses included in the next, final version of the manuscript.

We realize that it is difficult to revise to a specific deadline. In the interest of protecting the conceptual advance provided by the work, we recommend a revision within 3 months. Please discuss the revision progress ahead of this time with me if you require more time to complete the revisions, or if you have questions or comments regarding the revision (also by video chat).

1. A data availability section providing access to data deposited in public databases is missing (where applicable).
2. Your manuscript contains statistics and error bars based on n=2. Please use scatter plots in these cases.

You can submit the revision either as a Scientific Report or as a Research Article. For Scientific Reports, the revised manuscript can contain up to 5 main figures and 5 Expanded View figures, and it should not exceed 27000 characters. If the revision leads to a manuscript with more than 5 main figures it will be published as a Research Article. In this case the Results and Discussion section should be separate. If a Scientific Report is submitted, these sections have to be combined. This will help to shorten the manuscript text by eliminating some redundancy that is inevitable when discussing the same experiments twice. In either case, all materials and methods should be included in the main manuscript file.

3) We replaced Supplementary Information with Expanded View (EV) Figures and Tables that are collapsible/expandable online. A maximum of 5 EV Figures can be typeset. EV Figures should be cited as 'Figure EV1, Figure EV2' etc... in the text and their respective legends should be included in the main text after the legends of regular figures.

4) a .docx formatted letter INCLUDING the reviewers' reports and your detailed point-by-point responses to their comments. As part of the EMBO publication's Transparent Editorial Process, EMBO reports publishes online a Review Process File (RPF) to

accompany accepted manuscripts. This File will be published in conjunction with your paper and will include the referee reports, your point-by-point response and all pertinent correspondence relating to the manuscript.

<https://www.embopress.org/page/journal/14693178/authorguide#transparentprocess>

5) a complete author checklist, which you can download from our author guidelines

<https://www.embopress.org/page/journal/14693178/authorguide>. Please insert information in the checklist that is also reflected in the manuscript. The completed author checklist will also be part of the RPF.

6) Please note that all corresponding authors are required to supply an ORCID ID for their name upon submission of a revised manuscript (). Please find instructions on how to link your ORCID ID to your account in our manuscript tracking system in our Author guidelines

Additional information on source data and instruction on how to label the files are available:

<https://www.embopress.org/page/journal/14693178/authorguide#sourcedata>

9) Our journal encourages inclusion of *data citations in the reference list* to directly cite datasets that were re-used and obtained from public databases. Data citations in the article text are distinct from normal bibliographical citations and should directly link to the database records from which the data can be accessed. In the main text, data citations are formatted as follows: "Data ref: Smith et al, 2001" or "Data ref: NCBI Sequence Read Archive PRJNA342805, 2017". In the Reference list, data citations must be labeled with "[DATASET]". A data reference must provide the database name, accession number/identifiers and a resolvable link to the landing page from which the data can be accessed at the end of the reference. Further instructions are available at <http://www.embopress.org/page/journal/14693178/authorguide#referencesformat>

10) Regarding data quantification (see Figure Legends:

<https://www.embopress.org/page/journal/14693178/authorguide#figureformat>)

11) The journal requires a statement specifying whether or not authors have competing interests (defined as all potential or actual interests that could be perceived to influence the presentation or interpretation of an article). In case of competing

interests, this must be specified in your disclosure statement. Further information: <https://www.embopress.org/competing-interests>

12) Please also note our reference format:

13) All Materials and Methods need to be described in the main text using our 'Structured Methods' format, which is required for all research articles. According to this format, the Methods section includes a Reagents and Tools Table (listing key reagents, experimental models, software and relevant equipment and including their sources and relevant identifiers) followed by a Methods and Protocols section describing the methods using a step-by-step protocol format. The aim is to facilitate adoption of the methodologies across labs. More information on how to adhere to this format as well as a downloadable template (.docx) for the Reagents and Tools Table can be found in our author guidelines:

I look forward to seeing a revised version of your manuscript when it is ready. Please let me know if you have questions or comments regarding the revision.

Kind regards,

Deniz Senyilmaz Tiebe

Deniz Senyilmaz Tiebe, PhD
Scientific Editor
EMBO Reports

Referee #1:

Hansen et al were able to solve the structure of a long-awaited substrate bound E2P state which is relevant for the molecular understanding of P-type ATPases, specifically the E1 E2 transition. To this end they made use of a G4 insertion mutant, which in smFRET studies had already shown to get stalled in an intermediate [Ca]E2P state. The study is overall well conducted and merits publications. Below my comments on how the manuscript can be improved.

Main comments:

1. My main concern relays on the relevance of the solved structure and how it may represent a 'real' intermediate state. It is based on the analysis of the authors itself that on page 7: "However, the distance spanned by this linkers is almost 6. longer than in the [Ca]E1P-ADP and E2P states (Figure 4A,B,D) and without the G4 insert it would not reach without undergoing some level of conformational changes (Figure S4)". Late in the discussion it is discussed that due to this observation "the A-M1 linker must undergo strain in the wildtype [Ca]E2P form. The linker plays a crucial role in formation of a calcium exit pathway,..." which I assume is illustrated in Figure 5 as the transient E2P-outward-opne state. Hence, do the authors then assume that the state captured with this insertion mutant can actually not be adopted by the WT? If so I think this should be stated more clearly, for one that this structure cannot be adopted by the WT and that the conclusion are thus a projection of the findings obtained through this structure.

2. In contrast to the lower activity of the G4 mutant seen in detergent, the construct appears to not be active at all any longer when reconstituted in saposin. The author assume that saposins further stabilize the intermediate state entirely stalling the transporter in this conformation.

2a. While I agree with this assumption, I suggest the authors write (page 4) "This may indicate that the saposin nanodiscs stabilizes a particular state AND/OR RESTRICTS CERATIN CONFORMATIONAL CHANGES occurring in the ATPase reaction....".

2b. Can the authors also comment to which extend this observation might hamper a comparison to the smFRET studies, which I assume where conducted entirely in detergent where the protein clearly could still undergo the transport cycle? At least this should be mentioned and briefly discussed

2c. can the authors comment on why not then attempt to obtain a structure in detergent which would better resemble or allow a direct comparison to the smFRET studies? Was it tried and did not succeed? Latter could again be an indication that the protein

is indeed entirely stalled in this conformation

3. Suggest to improve here and there on how to better guide the reader. For example I know it is very challenging to display, describe and convey the movements of especially the cytosolic domains of P-type ATPase in a 2D format as manuscript figures. However, at several instances I believe it could be better done. Some examples:

- To differentiate between E1 and E2 states e.g. a nice hallmark is the proximity of the TGES motif to the phosphorylation site. This is nicely visible in Figure 3 but poorly used as a guide when describing the changes in the text. Could be added to the sentence on page 5 "Indeed, the cytosolic domains, although configured in an E2P-like conformation, are not yet tilted relative to the membrane plane, which is otherwise typical of.."
- perhaps mention why E2P state is usually ADP insensitive (aligns with the structures)"
- Figure 1 c-d could be improved. It is hard to dissect what the authors want to show. Perhaps try to show a separate panel for the TM only? From what is shown it is hard to tell to which the new structure resembles rather E1 or E2.
- Figure 2. Visually panel B looks more like C although the TM rather resembles an E1 state. Perhaps by highlighting and labelling some of the residues (coordinating residues or those that might pull or constrain a Calcium) described in the text might help here. Also is the E2P state an HE2P-occluded state or a E2P-outward-open state. I assume the former considering one of the main takes from the discussion. But then this should be better and more clearly stated throughout the text and not just be mentioned in the discussion.

4. While I get it, I would appreciate if the authors could explain and include it better in the manuscript why they created a homology model for the [Ca]E1P-ADP state and not used the one from SERCA. Would the study not benefit from also including a comparison to SERCA itself, especially in the discussion when extracting big picture take home messages from the study? At least that is real existing structure and that part of the transport cycle should still be comparable.

5. I would appreciate if more EM data could be shown. For example I lack any local resolution maps, or more model/maps figures with respective sigma. For example the position of Glu292 is greatly discussed and from figure 2 the density in that region in general appears generally rather poor. What is the local resolution here, for the entire helix and surrounding Glu292? How well does the density support this region in general? Also for other residues?

6. I was initially a bit puzzled when reading through the discussion where on page 9 differences to SERCA are mentioned (see sentence starting with "Unlike for SERCA...". I believe to have understood it better and it probably relates to SERCA likely adopting a stable E2P outward-open state capable of going reverse mode. However, this could be much better explained before, and even already in the introduction. This is also relevant to point 4 with regards to how relevant are these findings in understanding eukaryote homologs like SERCA.

Minor suggestions:

- Abstract line 3 missing space between 'several aspects'
- Page 5. Since the intermediate state does not entirely resemble a known state I would suggest to write accordingly, e.g. "Hence, the TM domain adopts RATHER an calcium-occluded E1P-like conformation, but the position of the M1-4 bundle is shifted relative"
- Page 5: "Altogether, these observations point to an intermediate state of the transition between the inward-occluded [Ca]E1P-ADP state to the outward-open E2P state, where ADP has been released,..." you rather mean the [Ca]E2P state no?
- Figure legend 2 and 3. PDB-ID 6ZFH should be 6ZHF
- Tone down Page 6 "The nucleotide binding site is empty; INDICATING THAT ADP must have been released prior to this point, probably at a preceding step going from the [Ca]E1P-ADP state to an ADP-sensitive [Ca]E1P state27."
- Page 8: However, the cytoplasmic headpiece of the three cytoplasmic domains is still not tilted into the configuration of the FINAL E2P state, but instead paused in the intermediate state due to..."
- Page 10: I assume you mean H+-binding here? "...presumably linked to very fast kinetics of occlusion,..."
- Page 10: "Hence the MOST LIKELY irreversible transition for LMCA1 is [Ca]E2P to [H]E2P, which completes the the A domain rotation transmitted by linker regions, and which includes calcium release with no outward-open Ca²⁺ intermediate explored per se."
- Figure 1A I believe in the scheme it should be [Ca]E1-ATP and not [Ca]E1P-ATP
- Figure 3 title: "Principal movements of cytosolic domains OF LMCA1 during the E1P to E2P transition"
- Figure 3 would highly benefit from labelling the Glu169/Arg491 in the figures, and also include the interaction distance right over the dotted line. Or keep it as it is and use an arrow. Perhaps it was just me but it took me a while to grasp it.
- Figure 4: I would also color at least the P domain as interaction to it are mentioned and relevant. Also in panel C have the section of the P domain colored in blue.
- Figure 5: label several components in the figure like A-M1 and A-M2 linker.

Referee #2:

The authors present a cryo-EM structure at 3.5 Å resolution of LMCA1 with a G4 extension on the M1 linker. The structure was determined under ATP turnover conditions, which was expected to trap a particular conformation with a bound calcium. The

intermediate is suggested to represent a new structure for Ca²⁺-ATPases and ion-transporting P2-type ATPases in general, and appears to be a hybrid between E1P and E2P states and a presumed intermediate in the E1P-E2P transition preceding the irreversible step of ion release.

1. Abstract: "severalsteps" should be "several steps"
2. There remains some concern about the effect of the saposin nanodiscs on LMCA1 function and the structural intermediate observed. Can the authors provide more convincing data or arguments to support their preconditions?
3. Table S1 does not state anywhere that the reported values are RMSDs.
4. It is difficult to discern from Figure 1C that the TM domain adopts a calcium-occluded E1P-like conformation.
5. In Figure 5, it would be helpful to have a reference point on each cytoplasmic domain such that a reader can get a visual sense of the movements involved in each transition.
6. A local resolution estimate (isosurface map colored according to local resolution) should be included. The RMSD comparisons and discussions of structural details (for example, Glu292) need to be put into context of the local resolution of these regions.
7. Figure S2, "pipline" should be pipeline

Referee #1:

Hansen et al were able to solve the structure of a long-awaited substrate bound E2P state which is relevant for the molecular understanding of P-type ATPases, specifically the E1 \leftrightarrow E2 transition. To this end they made use of a G4 insertion mutant, which in smFRET studies had already shown to get stalled in an intermediate [Ca]E2P state. The study is overall well conducted and merits publications. Below my comments on how the manuscript can be improved.

Main comments:

1. My main concern relates on the relevance of the solved structure and how it may represent a 'real' intermediate state. It is based on the analysis of the authors itself that on page 7: "However, the distance spanned by this linker is almost 6. longer than in the [Ca]E1P-ADP and E2P states (Figure 4A,B,D) and without the G4 insert it would not reach without undergoing some level of conformational changes (Figure S4)". Late in the discussion it is discussed that due to this observation "the A-M1 linker must undergo strain in the wildtype [Ca]E2P form. The linker plays a crucial role in formation of a calcium exit pathway,..." which I assume is illustrated in Figure 5 as the transient E2P-outward-open state. Hence, do the authors then assume that the state captured with this insertion mutant can actually not be adopted by the WT? If so I think this should be stated more clearly, for one that this structure cannot be adopted by the WT and that the conclusion are thus a projection of the findings obtained through this structure.

We thank for this comment. We have made a more thorough qualification of the point with a new analysis that shows that the A-M1 linker can span the gap in a physically plausible form, which however is also likely a strained conformation (Fig. EV4). In short, we remodelled the linker in "coot" allowing it to deviate from the local density that flanks the G₄-insert. This can shorten the gap by 2 Å, and a WT linker can reach. It supports the conclusion that the form captured by the G4 construct represents quite closely a WT state. Likely, the intermediate of the WT protein is strained and will relax upon Ca²⁺ release. We have added the following text:

"However, the distance spanned by this linker is almost 6Å longer than in the [Ca]E1P-ADP and E2P states (Figure 4A,B,D). The WT linker lacking the G₄ insert cannot span this gap without a conformational rearrangement. To assess if the WT linker can in principle link the configuration of domains in the [Ca]E2P intermediate state, we allowed it to deviate locally from the G₄-insert form (Figure EV4). This is not necessarily capturing the actual structure of the linker and M1 in the WT [Ca]E2P state, but it shows that the WT protein can assume such intermediate conformations in a physically realistic form. Likely it represents a strained intermediate state that relaxes upon calcium release"

Figure EV4: The A-M1 linker can span the domains in a strained conformation in WT LMCA1 in [Ca]E2P. (A) The break in linker A-M1 is indicated with numbered residues and the $C_{\alpha}(\text{Leu39}) - C_{\alpha}(\text{Pro46})$ distance is indicated. **(B)** Lys44 and Asp45 is connected without the G_4 insert by remodelling the loop in coot. The linker can span the domains in an extended conformation if a local deviation from the density is allowed. The $C_{\alpha}(\text{Leu39}) - C_{\alpha}(\text{Pro46})$ distance is measured like in Figure 4. The map is shown at $\sigma=0.225$.

2. In contrast to the lower activity of the G_4 mutant seen in detergent, the construct appears to not be active at all any longer when reconstituted in saposin. The author assume that saposins further stabilize the intermediate state entirely stalling the transporter in this conformation.

2a. While I agree with this assumption, I suggest the authors write (page 4) "This may indicate that the saposin nanodiscs stabilizes a particular state AND/OR RESTRICTS CERATIN CONFORMATIONAL CHANGES occurring in the ATPase reaction....".

Thank you for your suggestion. This is included in the text, where we now write in Results: *"This may indicate that the saposin nanodisc stabilizes a particular state and/or restricts certain conformational changes occurring in the ATPase reaction for LMCA1 and in particular associated with the G_4 construct. Knowing these preconditions, we proceeded with structural analysis of the sample."*

And later also in Discussion:

"...Therefore it accumulates in the [Ca]E2P state as revealed here by cryo-EM under ATPase turn-over conditions in a lipid nanodisc. At the same time, the nanodisc appears to block the ATPases activity of G_4 -LMCA1 (which shows an impaired, but significant activity in detergent) by stabilizing this particular state, although mechanisms for this cannot be detailed here due to a very low resolution of the map for saposin nanodisc features."

2b. Can the authors also comment to which extend this observation might hamper a comparison to the smFRET studies, which I assume where conducted entirely in detergent where the protein clearly could still undergo the transport cycle? At least this should be mentioned and briefly discussed

Indeed, our original intention was to perform also smFRET measurements in the saposin solubilized state. Unfortunately, this is not meaningful when the protein is stalled.

2c. can the authors comment on why not then attempt to obtain a structure in detergent which would better resemble or allow a direct comparison to the smFRET studies? Was it tried and did not succeed? Latter could again be an indication that the protein is indeed entirely stalled in this conformation

We tried to determine structures in detergent, but unsuccessfully. The 2D classes were poorly defined, which may indicate a very heterogenous and dynamic sample on grids that impaired cryo-EM procedures.

3. Suggest to improve here and there on how to better guide the reader. For example I know it is very challenging to display, describe and convey the movements of especially the cytosolic domains of P-type ATPase in a 2D format as manuscript figures. However, at several instances I believe it could be better done. Some examples:

- To differentiate between E1 and E2 states e.g. a nice hallmark is the proximity of the TGES motif to the phosphorylation site. This is nicely visible in Figure 3 but poorly used as a guide when describing the changes in the text. Could be added to the sentence on page 5 "Indeed, the cytosolic domains, although configured in an E2P-like conformation, are not yet tilted relative to the membrane plane, which is otherwise typical of.."

We thank for the suggestion. We now write:

"Indeed, the cytosolic domains, although configured in an E2P-like conformation with a much shorter distance between the TGES loop and the phosphorylation site when compared to the E1P conformation, are not yet tilted relative to the membrane plane, which is otherwise typical of the E2P state (Figure 1E).

- perhaps mention why E2P state is usually ADP insensitive (aligns with the structures)"

We now write (top of page 6):

"Visual inspection of the cryo-EM density map indicates no density for bound nucleotide at the N domain despite a 1 mM background of ATP/ADP in the cryo-EM sample. This indicates that the N domain adopts an ADP-insensitive conformation in progression of a forward transport cycle, a hallmark of E2P states."

- Figure 1 c-d could be improved. It is hard to dissect what the authors want to show. Perhaps try to show a separate panel for the TM only? From what is shown it is hard to tell to which the new structure resembles rather E1 or E2.

We thank for the feed-back. We have updated Figure 1 with zoom-in panels of both the TM domain and the cytosolic domains.

- Figure 2. Visually panel B looks more like C although the TM rather resembles an E1 state. Perhaps by highlighting and labelling some of the residues (coordinating residues or those that might pull or constrain a Calcium) described in the text might help here. Also is the E2P state an HE2P-occluded state or a E2P-outward-open state. I assume the former considering one of the main takes from the discussion. But then this should be better and more clearly stated throughout the text and not just be mentioned in the discussion.

M4 and M6 are in a closed conformation in all three structures. However, in the [Ca]E2P state, M1-4 are not rotated into a E2 position yet, and therefore we ascribe the TM to adopt an E1P-like conformation. We hope it is clearer after updating Figure 1. We have updated all labels on figures of the crystal structure of LMCA1 to [H]E2P to highlight that it is occluded.

4. While I get it, I would appreciate if the authors could explain and include it better in the manuscript why they created a homology model for the [Ca]E1P-ADP state and not used the one from SERCA. Would the study not benefit from also including a comparison to SERCA itself, especially in the discussion when extracting big picture take home messages from the study? At least that is real existing structure and that part of the transport cycle should still be comparable.

We have used a homology model of LMCA1 in our figures to show differences within the same protein, although we also refer to SERCA [Ca₂]E1-AIFx-ADP when appropriate (e.g. RMSD values in comparison of states are only computed between experimental structures of SERCA and LMCA1). Despite many similarities between SERCA and LMCA1, they do show critical differences, e.g. in the E2-BeFx forms, where SERCA adopts a stable outward-open E2P conformation, and LMCA1 adopts

a closed [H]E2P conformation, so in figures we prefer to pair with the homology model. We write on page 5:

Since the E2P states of LMCA1 are different from SERCA and crucial for our analysis, we have used the homology model of LMCA1 instead of the [Ca₂]E1-A1Fx-ADP structure of SERCA to visually present the conformational changes from [Ca]E1P-ADP to E2P within LMCA1."

We also discuss such difference in the Discussion

5. I would appreciate if more EM data could be shown. For example I lack any local resolution maps, or more model/maps figures with respective sigma. For example the position of Glu292 is greatly discussed and from figure 2 the density in that region in general appears generally rather poor. What is the local resolution here, for the entire helix and surrounding Glu292? How well does the density support this region in general? Also for other residues?

We have included panel D-F of the local resolution in Figure 2. Furthermore, the final structure in Figure S2 is changed for a local resolution map.

6. I was initially a bit puzzled when reading through the discussion where on page 9 differences to SERCA are mentioned (see sentence starting with "Unlike for SERCA...". I believe to have understood it better and it probably relates to SERCA likely adopting a stable E2P outward-open state capable of going reverse mode. However, this could be much better explained before, and even already in the introduction. This is also relevant to point 4 with regards to how relevant are these findings in understanding eukaryote homologs like SERCA.

We have included a section in the discussion that discusses a possible difference in mechanism of SERCA:

"Irreversibility for SERCA

All P-type ATPases require an irreversible step to transport against the concentration gradients. However, G₄-SERCA is prone to reverse reactions with BeFx and Ca²⁺ to form a [Ca₂]E2-BeFx form¹⁶, and even Pi and Ca²⁺ can reverse it into the [Ca₂]E2P state¹⁷. This suggests that the transition from [Ca]E2P to E2P is not an irreversible step for SERCA. In SERCA, the outward-open E2-BeFx structure reveals that the A domain is not yet fully rotated¹⁴. This suggests that when the ion binding site remains accessible in this outward-open state, the nucleotide binding might still require protection from a reverse ADP reaction, provided by the TGES loop in the A domain.

Probably, the irreversible step of the SERCA cycle is the proton-occluding closure of the extracellular pathway, where the A domain completes its rotation, dephosphorylation takes place, and the cycle proceeds with modulatory ATP³⁴. The final rotation coupled to proton occlusion allows the glutamate in the TGES loop to dephosphorylate the phosphorylation site. Molecular dynamics (MD) simulations and single-molecule studies could be informative approaches for future studies into these critical partial reactions."

Minor suggestions:

- Abstract line 3 missing space between 'several aspects'. **Changed**

- Page 5. Since the intermediate state does not entirely resemble a known state I would suggest to write accordingly, e.g. "Hence, the TM domain adopts RATHER an calcium-occluded E1P-like conformation, but the position of the M1-4 bundle is shifted relative"

We have changed the sentence to " Hence, the TM domain appears to adopt a calcium-occluded E1P-like conformation, where M1-2 are not rotated into the E2P conformation yet (Figure 1C). "

- Page 5: "Altogether, these observations point to an intermediate state of the transition between the inward-occluded [Ca]E1P-ADP state to the outward-open E2P state, where ADP has been released,..." you rather mean the [Ca]E2P state no?

We mean that the structure represents an intermediate for the overall transition, and now write "Altogether, these observations point to the G₄-LMCA1 adopting an intermediate state of the transition between the inward-occluded [Ca]E1P-ADP state and the outward-open E2P state, where ADP has been released and calcium is still bound."

- Figure legend 2 and 3. PDB-ID 6ZFH should be 6ZHF. Thank you for pointing this out. It is now changed.

- Tone down Page 6 "The nucleotide binding site is empty; INDICATING THAT ADP must have been released prior to this point, probably at a preceding step going from the [Ca]E1P-ADP state to an ADP-sensitive [Ca]E1P state²⁷." *We fully agree, changed.*

- Page 8: However, the cytoplasmic headpiece of the three cytoplasmic domains is still not tilted into the configuration of the FINAL E2P state, but instead paused in the intermediate state due to..." *Included, we now write (top of page 9): "However, the cytoplasmic headpiece of the three cytoplasmic domains is still not tilted into the configuration of the calcium-released E2P state, but rather paused in the intermediate state due to the G₄-extension of the A-M1 linker that relaxes strain."*

- Page 10: I assume you mean H⁺-binding here? "...presumably linked to very fast kinetics of occlusion,..."

Yes, and we now write: "...and presumably it is linked to very fast kinetics of proton occlusion, as described above, and the subsequent dephosphorylation reaction of LMCA1 once the [H]E2P state is reached"

- Page 10: "Hence the MOST LIKELY irreversible transition for LMCA1 is [Ca]E2P to [H]E2P, which completes the the A domain rotation transmitted by linker regions, and which includes calcium release with no outward-open Ca²⁺ intermediate explored per se."

Agree, we have included that (page 11, top)

- Figure 1A I believe in the scheme it should be [Ca]E1-ATP and not [Ca]E1P-ATP. *Indeed - it is now changed.*

- Figure 3 title: "Principal movements of cytosolic domains OF LMCA1 during the E1P to E2P transition" *Included.*

- Figure 3 would highly benefit from labelling the Glu169/Arg491 in the figures, and also include the interaction distance right over the dotted line. Or keep it as it is and use an arrow. Perhaps it was just me but it took me a while to grasp it.

We thank for the suggestion. We made that addition.

- Figure 4: I would also color at least the P domain as interaction to it are mentioned and relevant. Also in panel C have the section of the P domain colored in blue. *We have updated the figure.*

- Figure 5: label several components in the figure like A-M1 and A-M2 linker. *The figure is updated*

Referee #2:

The authors present a cryo-EM structure at 3.5 Å resolution of LMCA1 with a G4 extension on the M1 linker. The structure was determined under ATP turnover conditions, which was expected to trap a particular conformation with a bound calcium. The intermediate is suggested to represent a new structure for Ca²⁺-ATPases and ion-transporting P2-type ATPases in general, and appears to

be a hybrid between E1P and E2P states and a presumed intermediate in the E1P-E2P transition preceding the irreversible step of ion release.

1. Abstract: "severalsteps" should be "several steps" **Changed**
2. There remains some concern about the effect of the saposin nanodiscs on LMCA1 function and the structural intermediate observed. Can the authors provide more convincing data or arguments to support their preconditions? **We have updated Figure S4 to show that the A-M1 linker can be connected in a physically realistic way, where WT still adopts the same overall conformation, only with a linker that is likely strained (see also response to reviewer 1).**
3. Table S1 does not state anywhere that the reported values are RMSDs. **It is now included**
4. It is difficult to discern from Figure 1C that the TM domain adopts a calcium-occluded E1P-like conformation. **The figure has been updated to show this point clearer**
5. In Figure 5, it would be helpful to have a reference point on each cytoplasmic domain such that a reader can get a visual sense of the movements involved in each transition. **We thank for the comments, and the figure is updated**
6. A local resolution estimate (isosurface map colored according to local resolution) should be included. The RMSD comparisons and discussions of structural details (for example, Glu292) need to be put into context of the local resolution of these regions. **We have included panel D-F of the local resolution in figure 2.**
7. Figure S2, "pipline" should be pipeline. **Corrected**

Dear Poul,

Thank you for submitting your revised manuscript. It has now been seen by one of the original referees.

As you can see, the referee finds that the study is significantly improved during revision and recommend publication. However, I need you to address the points below before I can accept the manuscript.

- We note that minor suggestion of referee #1 may improve the presentation of the study, but we leave it up to you to address it.
- We believe that the format of the manuscript is better suited for our Reports format rather than Scientific Article, which needs to be updated accordingly during the resubmission - i.e. by combining the Results and the Discussions sections. Please see <https://www.embopress.org/page/journal/14693178/authorguide#researcharticleguide>
- Please remove the figures from the manuscript text file and place the figure legends at the very end of the text. Also, Acknowledgments section needs to be placed after the Data Availability section.
- Please provide 3-5 keywords for your study. These will be visible in the html version of the paper and on PubMed and will help increase the discoverability of your work.
- We note that the links provided in the Data Availability section only resolve to the homepage of the respective databases. Please provide links that directly resolve to the datasets 9GQO and EMD-51510.
- Please add a Disclosure Statement & Competing Interests section to the manuscript (please see <https://www.embopress.org/page/journal/14693178/authorguide#conflictsofinterest>).
- As per our format requirements, in the reference list, citations should be listed in alphabetical order and then chronologically, with the authors' surnames and initials inverted; where there are more than 10 authors on a paper, 10 will be listed, followed by 'et al.'. Please see <https://www.embopress.org/page/journal/14693178/authorguide#referencesformat>
- We note that D87 and D88 cells of the Author Checklist have not been filled in.
- Reused publicly available data need to be cited in the form of data citations in the reference list in addition to the publication containing the dataset - e.g. 1T5T, 6ZHF, 15t5, 6zhh, 3B9B, 6ZHG, 39BR. Data references should include authors, when possible, the year, the full name of the database where the data is available, the accession number or DOI and, importantly, a resolvable link that points directly to the dataset. The resolvable link can be in the form of either a plain URL, a DOI or an identifiers.org construct. I paste an example below:

In-text: "...were grouped based on the relative levels of AR-Vs expressed, mainly AR-V7 (Hörnberg et al, 2011; Data ref: Hörnberg et al, 2011)."

Reference list:

Hörnberg E, Ylitalo EB, Crnalic S, Antti H, Stattin P, Widmark A, Bergh A, Wikström P (2011) Gene Expression Omnibus GSE29650 (<https://www.ncbi.nlm.nih.gov/geo/query/acc.cgi?acc=GSE29650>). [DATASET]

Hörnberg E, Ylitalo EB, Crnalic S, Antti H, Stattin P, Widmark A, Bergh A, Wikström P (2011) Expression of androgen receptor splice variants in prostate cancer bone metastases is associated with castration-resistance and short survival. PLoS One 6: e19059

Please see <https://www.embopress.org/page/journal/14693178/authorguide#referencesformat> for further information.

- We note that the funding information is incomplete in our manuscript tracking system - the Novo Nordisk Foundation (ICE-T facility, grant no. NNF20OC0060483) is currently missing.
- We note that majority of the figures are in Photoshop format. For publication, we require TIFF, PDF or EPS files in PC or Macintosh format, preferably from PhotoShop or Illustrator software. We cannot accept Freehand, Canvas, CorelDRAW or MacDrawPro files. These files must be converted to postscript (eps) format. For any figures submitted in Photoshop or TIF(F) format we require layered files to be sent whereby all text, arrows or additional attributes are placed on individual layers within the file. For line art/charts/graphs we prefer to work with Adobe Illustrator AI, EPS, or high-resolution PDF files.
- We note that Figure 3C is currently not called out in the text. An appendix Table S1 is not provided, but called out in the manuscript text.
- Table EV1 and its legend need to be removed from the manuscript and uploaded as a separate Expanded View file.
- We note that movies were referred to in the text, but no movies were provided.
- All research articles submitted as revised versions must include a structured methods section that includes a Reagents and Tools Table followed by a Methods and Protocols section. Please see <https://www.embopress.org/page/journal/14693178/authorguide#structuredmethods> for further information.
- Please fill the attached source data checklist with the accession information of the deposited cryo-EM data.
- Materials and methods should be renamed as Methods.
- The manuscript sections should be in the following order: Title page - Abstract & Keywords - Introduction - Results - Discussion - Methods - Data Availability - Acknowledgments - Disclosure Statement & Competing Interests - References - Figure Legends - (Main Tables with legends if applicable) - Expanded View Figure Legends.
- Our production/data editors have asked you to clarify several points in the figure legends:

- o Please note that information related to n is missing in the legend of figure EV 1b.
- o Please note that the error bars are not defined in the legend of figure EV 1b.
- Papers published in EMBO Reports include a 'synopsis' and 'bullet points' to further enhance discoverability. Both are displayed on the html version of the paper and are freely accessible to all readers. The synopsis includes a short standfirst summarizing the study in 1 or 2 sentences (max 35 words) that summarize the paper and are provided by the authors and streamlined by the handling editor. I would therefore ask you to include your synopsis blurb and 3-5 bullet points listing the key experimental findings.
- In addition, please provide an image for the synopsis. This image should provide a rapid overview of the question addressed in the study but still needs to be kept fairly modest since the image size cannot exceed 550 (width) x 300-600 (height) pixels.

Thank you again for giving us to consider your manuscript for EMBO Reports, I look forward to your minor revision.

Kind regards,

Deniz

--

Deniz Senyilmaz Tiebe, PhD
Senior Scientific Editor
EMBO Reports

Referee #1:

I thank the authors for addressing my comments in a satisfactory manner.
The manuscript merits submission and i congratulate the authors.
One minor suggestions: Figure 2 panel D-F should be moved to supplementaries.

All editorial and formatting issues were resolved by the authors.

Prof. Poul Nissen
Aarhus University
Dept. Molecular Biology and Genetics
Universitetsbyen 81, bld. 1874
Aarhus DK-8000
Denmark

Dear Poul,

Thank you for submitting your revised manuscript. I have now looked at everything and all is fine. Therefore, I am very pleased to accept your manuscript for publication in EMBO Reports.

Congratulations on a nice work!

Kind regards,

Deniz Senyilmaz Tiebe

--

Deniz Senyilmaz Tiebe, PhD
Senior Scientific Editor
EMBO Reports

--
